# State-dependent activity dynamics of hypothalamic stress effector neurons

Aoi Ichiyama[1], Samuel Mestern[1], Gabriel B Benigno[2,3], Kaela E Scott[1,4], Brian L Allman[4], Lyle Muller[2,3,5]*[†], Wataru Inoue[5,6]*[†]

[1]Graduate Program in Neuroscience, Western University, London, Canada; [2]Department of Mathematics, Western University, London, Canada; [3]Brain and Mind Institute, Western University, London, Canada; [4]Department of Anatomy and Cell Biology, Schulich School of Medicine and Dentistry, Western University, London, Canada; [5]Robarts Research Institute, Western University, London, Canada; [6]Department of Physiology and Pharmacology, Schulich School of Medicine and Dentistry, Western University, London, Canada

**Abstract** The stress response necessitates an immediate boost in vital physiological functions from their homeostatic operation to an elevated emergency response. However, the neural mechanisms underlying this state-dependent change remain largely unknown. Using a combination of in vivo and ex vivo electrophysiology with computational modeling, we report that corticotropin releasing hormone (CRH) neurons in the paraventricular nucleus of the hypothalamus (PVN), the effector neurons of hormonal stress response, rapidly transition between distinct activity states through recurrent inhibition. Specifically, in vivo optrode recording shows that under non-stress conditions, $CRH_{PVN}$ neurons often fire with rhythmic brief bursts (RB), which, somewhat counterintuitively, constrains firing rate due to long (~2 s) interburst intervals. Stressful stimuli rapidly switch RB to continuous single spiking (SS), permitting a large increase in firing rate. A spiking network model shows that recurrent inhibition can control this activity-state switch, and more broadly the gain of spiking responses to excitatory inputs. In biological $CRH_{PVN}$ neurons ex vivo, the injection of whole-cell currents derived from our computational model recreates the in vivo-like switch between RB and SS, providing direct evidence that physiologically relevant network inputs enable state-dependent computation in single neurons. Together, we present a novel mechanism for state-dependent activity dynamics in $CRH_{PVN}$ neurons.

*For correspondence:
lmuller2@uwo.ca (LM);
winoue@robarts.ca (WI)

[†]Co-senior authors.

**Competing interest:** The authors declare that no competing interests exist.

## Editor's evaluation

This is the first electrophysiological study showing high-quality spike activity in vivo from identified CRH neurons in the hypothalamic paraventricular nucleus. The authors make the surprising observation that CRH neurons exhibit brief high-frequency rhythmic bursts in unstressed mice that are converted to a sustained, single spike firing mode by an acute stressor. Their findings are supported by a computational model that suggests that feedback inhibition may regulate the activity patterns of CRH neurons in distinct states. The work provides a framework for new studies exploring firing characteristics in discrete physiological and emotional states in the hypothalamus and perhaps other regions.

## Introduction

Mounting an optimal stress response is critical for survival. This requires neural mechanisms that effectively switches vital physiological functions between homeostatic operation and emergency response.

For example, the activation of the hypothalamic-pituitary-adrenal (HPA) axis elevates the systemic levels of glucocorticoids, a hallmark of stress. Under non-stress conditions, glucocorticoids are also produced, albeit at lower levels, for their widespread physiological actions (*de Kloet et al., 2005*; *Lightman, 2008*). While HPA axis (patho)physiology has been extensively studied with relevance to stress-related disorders (*Gold and Chrousos, 1999*; *McEwen, 2005*), we know surprisingly little about the basic neural mechanisms that powerfully and bidirectionally transition HPA axis activity from baseline to levels of elevated stress.

The brain ultimately drives the HPA axis by exciting a group of neuroendocrine neurons that synthesize corticotropin-releasing hormone (CRH). The cell bodies of CRH neurons reside in the paraventricular nucleus of the hypothalamus (PVN) and project their axons to the median eminence. At the median eminence, axonally released CRH enters the hypophyseal portal circulation and initiates the downstream hormonal cascades (*Swanson and Sawchenko, 1980*). Numerous studies have shown that various acute stressors induce de novo expression of immediate early genes in CRH$_{PVN}$ neurons, establishing their stress-induced activity increase on the time scale of minutes to hours (*Cole and Sawchenko, 2002*; *Itoi et al., 2004*; *Matovic et al., 2020*; *Wamsteeker Cusulin et al., 2013*). Recent in vivo fiber photometry imaging revealed highly dynamic activity changes where population Ca$^{2+}$ signals in CRH$_{PVN}$ neurons (a proxy for population activity) increase rapidly (within seconds) and reversibly in response to repetitive, short-aversive stimuli (*Daviu et al., 2020*; *Kim et al., 2019b*; *Kim et al., 2019a*; *Yuan et al., 2019*).

How do individual CRH$_{PVN}$ neurons fire action potentials during the rapid transitions between baseline and stress conditions? Although the in vivo firing activity of single units in the PVN was first studied more than 30 years ago (*Hamamura et al., 1986*; *Kannan et al., 1987*; *Saphier and Feldman, 1985*; *Saphier and Feldman, 1988*; *Saphier and Feldman, 1991*), these pioneering experiments lacked neurochemical identification; *i.e.*, it could not be confirmed whether the spiking activity was of CRH neurons or another sub-population of neurons within the PVN. This is because in vivo extracellular electrophysiology is inherently blinded to neuronal identities, and CRH neurons are intermingled with a mosaic of neuron-types deep in the hypothalamus (*Simmons and Swanson, 2009*). Thus, despite decades of research on CRH$_{PVN}$ neurons, their firing activities in vivo have remained inaccessible. This is an important gap in our knowledge because single-neuron firing patterns result from dynamic interaction between intrinsic properties and network dynamics (*Destexhe et al., 2001*; *Herz et al., 2006*; *Leng and MacGregor, 2018*), and hence will provide essential clues to probe how CRH$_{PVN}$ neurons—the gatekeepers of stress outputs—process the converging brain commands during baseline conditions and in response to stress.

Here we have implemented in vivo single-unit extracellular recording of optogenetically identified CRH$_{PVN}$ neurons in mice, and subsequently developed a network model that explains in vivo state-dependent firing dynamics changes. For the first time, we report that CRH$_{PVN}$ neurons fire in two distinct modes: rhythmic brief bursts (RB) and single spikes (SS) of random spike intervals. Somewhat counterintuitively, RB occurred under the baseline conditions and constrained the overall firing rate at low levels due to long (~2 s) interburst silence. On the other hand, sustained SS underlies stress-induced increase in firing. Using tight combinations of experimental and computational approaches, we show that a recurrent inhibitory network generates RB and constrains the overall firing rate due to negative feedback inhibition. A spiking network model showed that release from recurrent inhibition upon stress stimuli permits continuous SS that increases the overall firing rate, thereby revealing a novel mechanism that enables efficient bidirectional controls of stress output neurons. To validate the spiking network model, we used ex vivo patch-clamp electrophysiology and injected excitatory and inhibitory inputs received by a cell in the computational model into actual biological neurons. CRH$_{PVN}$ neurons in these experiments exhibited in vivo-like bursting behavior when driven by the network current, but not when driven by simple step-current stimuli. Ultimately, our collective results identified a novel network mechanism that underlies state-dependent firing activity dynamics at the effector neurons of hormonal stress response.

## Results

### Identifying CRH neurons in vivo

To examine firing patterns of individual $CRH_{PVN}$ neurons in vivo, we used optrode recording (*Lima et al., 2009*). Briefly, we first 'tagged' CRH neurons by expressing an excitatory opsin channelrhodopsin2 (ChR2) using adeno-associated virus (AAV) carrying a cre-dependent promoter construct injected into the PVN of CRH-Ires-Cre mice crossed with Ai14 td-tomato reporter line (*Wamsteeker Cusulin et al., 2013*). This resulted in ChR2-EYFP expression in $CRH_{PVN}$ neurons that express td-tomato (*Figure 1A*), similar to recent studies using the same approach (*Bittar et al., 2019*; *Füzesi et al., 2016*; *Kim et al., 2019a*; *Wamsteeker Cusulin et al., 2013*; *Yuan et al., 2019*). Important for the desired PVN neuron identification, these ChR2-expressing neurons could now be detected on the basis of their light-induced spike firing. We isolated 36 single units in the PVN area from 10 adult male mice under urethane anaesthesia. Among these preisolated single units, 18 neurons (from 10 mice) were 'light-responsive'; *i.e.*, they fired action potentials in response to pulses of blue light (5 ms, $\lambda$ =465 nm) with short latency (7.2 ± 2.6 ms, n=18; *Figure 1B and C*). This responsiveness to light had a binary effect (*Figure 1D*): the light-responsive neurons reliably increased their probability of firing (7.1 ± 6.3% preonset to 66.3 ± 21.3% postonset, n=18), whereas the light-non-responsive neurons did not (3.3 ± 3.1% preonset to 2.0 ± 2.8% postonset, n=18). Furthermore, in response to longer light pulses (50 ms), the light-responsive neurons fired a train of action potential with frequency adaptation (*Figure 1E and F*). Consequently, these light-responsive neurons were defined as $CRH_{PVN}$ neurons.

### $CRH_{PVN}$ neurons can fire in two distinct modes

We found that, under the baseline (non-stress) conditions, many $CRH_{PVN}$ neurons occasionally fired in a distinct brief train of high-frequency (>100 Hz) bursts, in addition to spikes with variable interspike intervals (ISI, *Figure 2A*). *Figure 2B* shows an ISI histogram of a representative neuron that demonstrated a bimodal distribution with a sharp peak around 2–10 ms (burst) and another wider peak around 100 ms (SS). Burst trains typically showed the shortest ISI at the start and followed by a few spikes with slightly increased ISIs. Thus, to quantify these brief burst firing, we set a criterion for burst detection with ISI ≤ 6 ms for the start of a burst, and subsequent spikes were considered part of the burst train as long as ISI remained below 20 ms. Using this burst-detection criterion, we found that the majority of $CRH_{PVN}$ neurons fired a brief burst at least once during the baseline (16 out of 18), but the burst rate was variable among $CRH_{PVN}$ neurons (*Figure 2C*). *Figure 2—figure supplement 1* shows ISI histograms for all units. For the analysis of burst properties below, we used neurons that showed at least 60 bursts during 10 min baseline recording (>0.1 bursts/s, purple circles in *Figure 2C*, n=10). Among these bursting $CRH_{PVN}$ neurons, each individual burst episode was brief, on average 3.1 ± 0.5 spikes, ranging between 2 and 6 spikes (*Figure 2D*). These brief bursts occurred at slow rhythms (rhythmic brief burst, RB), intervened by long, mostly silent, interburst intervals (IBI, 1.8 ± 0.6 s, n=10, *Figure 2E*).

### $CRH_{PVN}$ neurons are constrained to low activity during rhythmic bursting

In many neurons, burst firing is postulated to carry behaviorally relevant information, representing specific network states different from that represented by SS of variable frequencies (*Krahe and Gabbiani, 2004*; *Steriade et al., 1993*). The classic role of $CRH_{PVN}$ neurons is to respond to stress stimuli with an increase in firing activity, driving the neuroendocrine release of CRH (*Ulrich-Lai and Herman, 2009*). Thus, we examined potential roles of RB in $CRH_{PVN}$ neurons' response to stress. To this end, we used electric stimulation of the sciatic nerve to model noxious sensory stimuli, as this approach has been shown to effectively elicit time-locked firing responses in median eminence-projecting parvocellular PVN neurons (*Day et al., 1985*; *Hamamura et al., 1986*; *Kannan et al., 1987*; *Saphier, 1989*; *Saphier and Feldman, 1985*; *Saphier and Feldman, 1991*) as well as to increase the circulating ACTH levels, indicative of HPA axis activation (*Feldman et al., 1981*), in anesthetized rats. For this experiment, we recorded 13 $CRH_{PVN}$ neurons (among 18 neurons recorded for the baseline firing analysis described above) that remained stable until the end of nerve stimulation session (see *Experimental Design* in Materials & Methods). Consistent with the earlier 'blinded' recordings (*Day et al., 1985*; *Hamamura et al., 1986*; *Kannan et al., 1987*; *Saphier, 1989*; *Saphier and Feldman,*

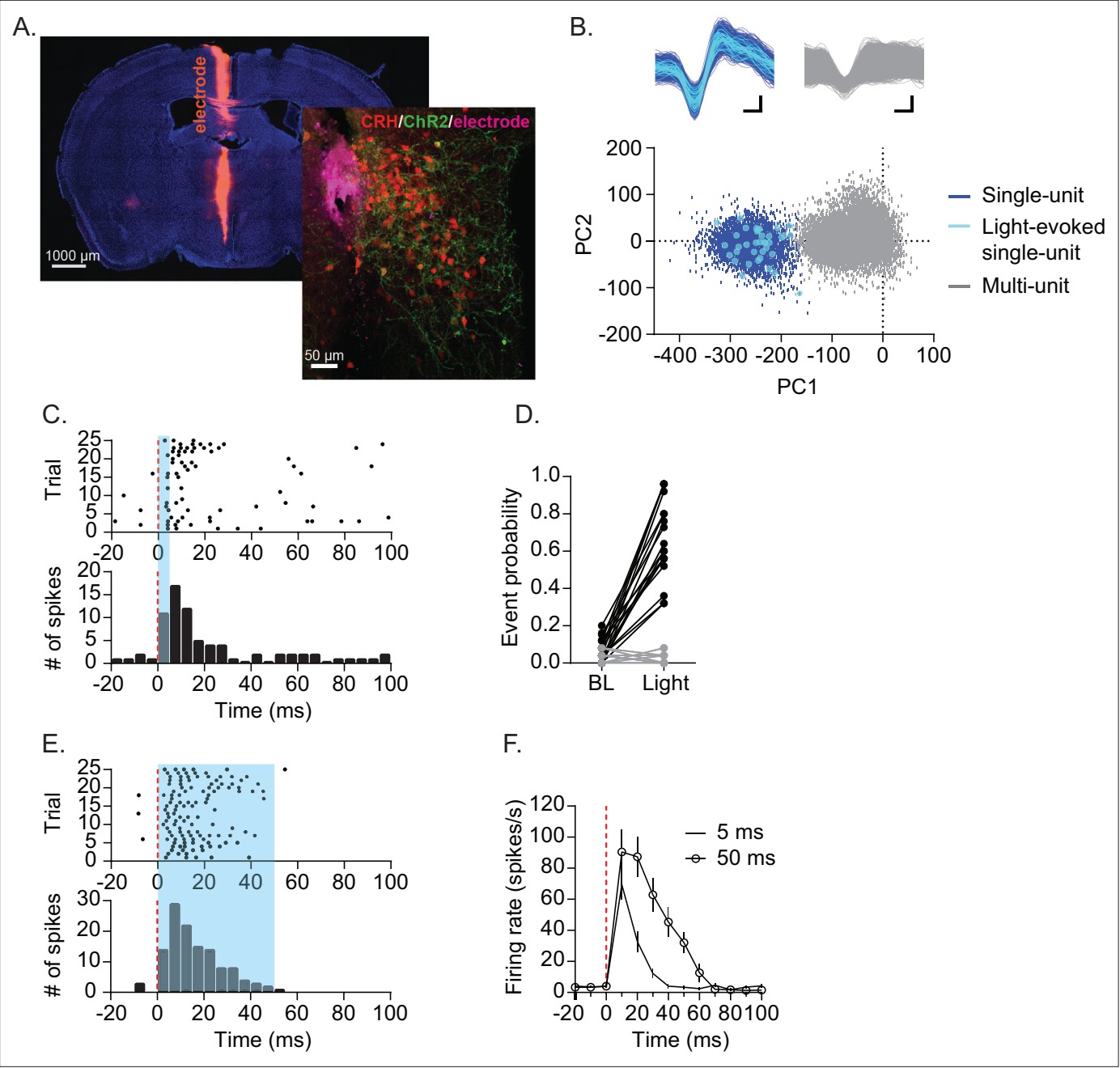

**Figure 1.** Optogenetic identification of CRH_PVN neuron single-unit. (**A**) Electrode tract (purple), TdTomato-expressing CRH_PVN neurons (red) and ChR2-EYFP (green) expression. (**B**) An example for an isolated single-unit (blue) that also responded to light (cyan). (**C**) Raster plot (top) and peristimulus time histogram (PSTH, bottom) in response to 5 ms blue light from a representative unit. (**D**) A summary graph for the probability of firing before (−20–0 ms) and after blue light illumination (0–20 ms). Light-responsive units in black (n=18) and non-responsive units in gray (n=18). (**E**) Raster plot (top) and PSTH (bottom) for a representative single-unit responding to 50 ms blue light. (**F**) Summary of firing rate time course following 5 ms (tick) and 50 ms (open circle) blue light. n=18. SD is represented as error bars.

The online version of this article includes the following source data for figure 1:

**Source data 1.** Optogenetic identification of CRHPVN neuron single-unit.

*1985*; *Saphier and Feldman, 1991*), sciatic nerve stimulation (1.6 mA) increased the firing rate (time averaged spike number) of CRH_PVN neurons (BL 3.063 ± 1.646 Hz vs NS 4.900 ± 1.962 Hz, paired $t$-test, p=0.0108, n=13; *Figure 3A, B and G*). Somewhat counter-intuitively, however, this activity increase was paralleled by a striking loss of RB (BL 0.1406 ± 0.1375 Hz vs NS 0.0474 ± 0.0410 Hz, paired $t$-test,

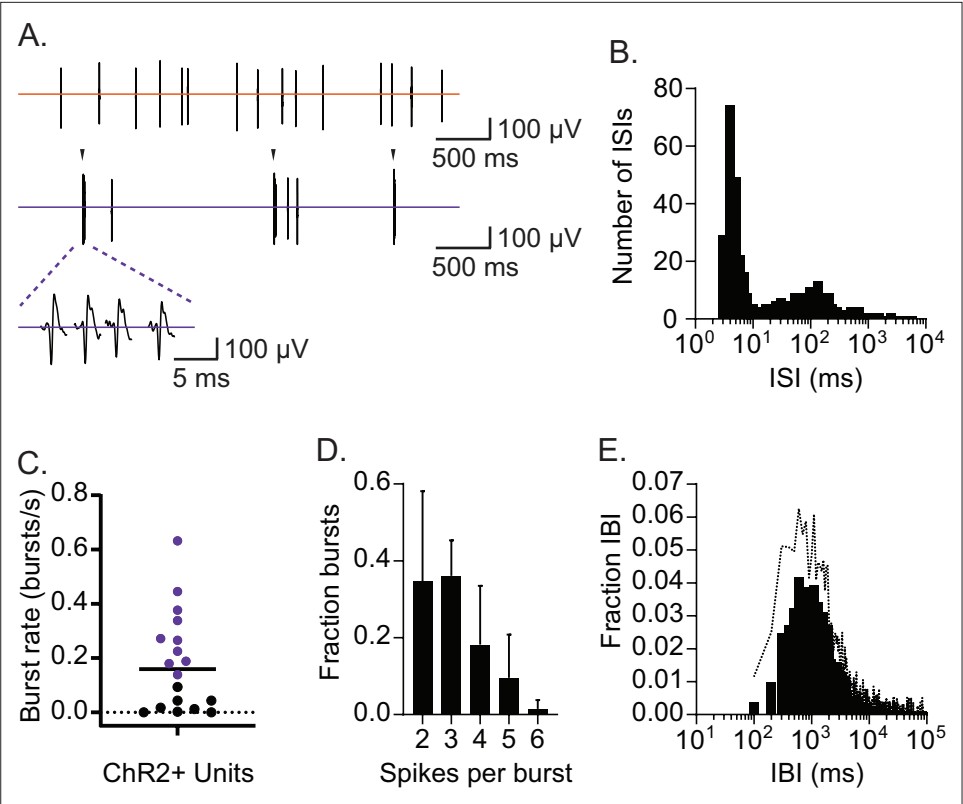

**Figure 2.** CRH$_{PVN}$ neurons fire in rhythmic bursts and single spikes. (**A**) Two distinct spike patterns in a representative single-unit (top: single spikes; bottom: bursts indicated by arrow). (**B**) Interspike interval (ISI) distribution for the representative single-unit shown in A. (**C**) A summary plot for burst rates among CRH$_{PVN}$ neurons during non-stress baseline recordings. The criterion for burst firing neurons (purple) was set as 0.1 burst/s. A horizontal bar indicates the average. (**D**) A summary graph for burst length distribution (n=10). (**E**) Summary interburst interval (IBI) distribution (n=10). Standard deviation is represented in the graphs as the error bars (**D**) and dotted line (**E**).

The online version of this article includes the following source data and figure supplement(s) for figure 2:

**Source data 1.** CRHPVN neurons fire in rhythmic bursts and single spikes.

**Figure supplement 1.** Interspike interval (ISI) histograms for all CRH$_{PVN}$ units.

---

p=0.0167, n=13, *Figure 3C, E and H*), and the firing rate increased due to SS (BL 2.702 ± 1.428 Hz vs NS 4.782 ± 1.987 Hz, paired *t*-test, p=0.0025, n=13; *Figure 3D, F*). In a representative case, we found that the firing pattern change lasted longer with higher intensity stimulation. *Figure 3—figure supplement 1* shows that, with the highest intensity stimulation (2 mA), the firing activity change did not completely return to the baseline within the trial duration of 15 s, resulting in continuous decrease of RB and increase of SS during the stimulation period. This finding suggests that RB paradoxically informs a certain 'low activity state' of CRH$_{PVN}$ neurons, whereas the 'high activity state', induced by stress, is due to SS at high rates.

Under baseline conditions, CRH$_{PVN}$ neurons showed spontaneous low-level firing activities; an observation in line with recent in vivo two-photon Ca$^{2+}$ imaging of CRH neurons in zebra fish larvae (*Vom Berg-Maurer et al., 2016*). We also found that some of CRH$_{PVN}$ neurons show time-dependent fluctuations in their firing rate with spontaneous and transient increase (*Figure 3J*). Thus, we next asked whether spontaneous 'high activity state' is also due to an increase in SS paralleled by a loss of RB. *Figure 3J* plots the running average firing rate (10 s bins) of one representative neuron showing spontaneous emergence of high activity states. We next overlaid the time course of individual spike's ISI to visualize the temporal relationship between the overall activity states (i.e. firing rate) and specific firing patterns (i.e. RB vs SS). Similar to the stress-induced activity increase, a loss of RB temporally correlated with the emergence of the high activity state (*Figure 3K*). On the other

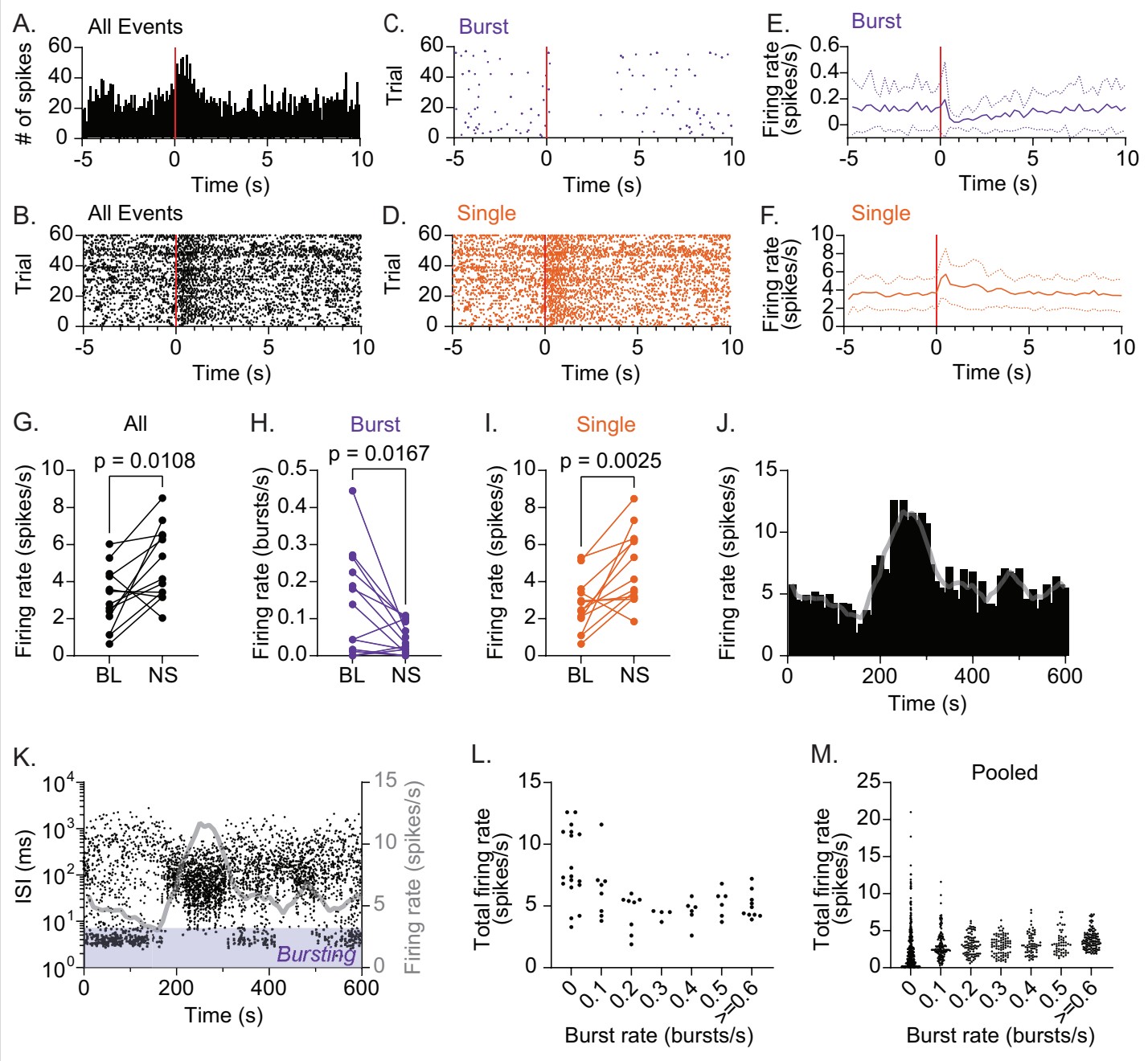

**Figure 3.** CRH$_{PVN}$ neurons are constrained to low activity during rhythmic bursting. (**A**) Peristimulus time histogram (PSTH) for a representative single-unit responding to sciatic nerve stimulation (1.5 mA, 0.5 ms × 5 pulses at 20 Hz, red line). (**B–D**) Raster plots for the unit shown in A for all spikes (**B**), bursts (**C**) and single spikes (SS) (**D**). (**E, F**) Summary time course for bursts (**E**) and SS (**F**). n=13. (**G–I**) Summary changes in all spikes (**G**, paired *t*-test, n=13, p=0.0108), bursts (**H**, paired *t*-test, n=13, p=0.0167) and SS (**I**, paired *t*-test, n=13, p=0.0025) before (baseline, BL) and after sciatic nerve stimulation (NS). (**J, K**) Time course of firing rate (**J**) and interspike interval (ISI) (**K**) for a representative single-unit during baseline recording. Gray lines are the running average (30 s) of firing rate. (**L**) The relationship between burst rate and total firing rate for the representative single-unit shown in J and K. For each time bin (10 s), total spike rate was plotted against burst rate. (**M**) Pooled data for all units (n=18).

The online version of this article includes the following source data and figure supplement(s) for figure 3:

**Source data 1.** CRHPVN neurons are constrained to low activity during rhythmic bursting.

**Figure supplement 1.** Firing pattern changes persist longer with higher intensity nerve stimulations.

hand, when CRH$_{PVN}$ neurons fired mainly with RB, their overall firing rate remained at relatively low levels. Intuitively, this is because of the brevity of burst episodes with mostly silent IBIs (*Figure 2A and E*), precluding the high rate of SS. *Figure 3L and M* plot the relationship between burst and total spike rate for every 10 s time bin for this unit, as well as for all units pooled (n=18), respectively. This analysis revealed that high levels of total spike rate only emerged when burst rate was at or near zero. Thus, our data suggest that a loss of RB permits CRH$_{PVN}$ neurons to increase firing rate with high rate of SS. In other words, the network activity state that drives RB constrains the activity of CRH$_{PVN}$ neurons at low levels.

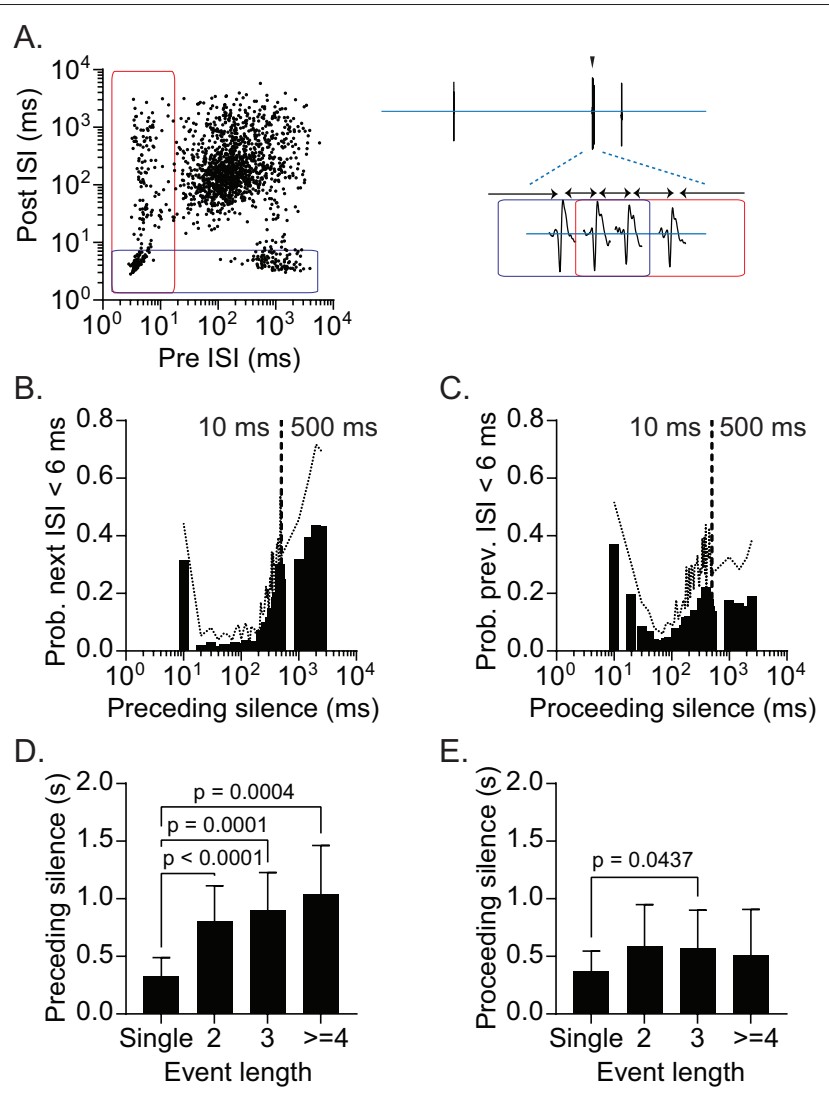

**Figure 4.** Prolonged silent periods precede burst firing. (**A**) Preceding and proceeding interspike interval (ISI) plotted for individual spikes recorded from a representative single-unit. Blue rectangle indicates spikes with proceeding ISI <6 ms. Red rectangle indicates spikes with preceding ISI <20 ms. Left. An example of a burst episode. (**B**) Summary of probability of burst firing relative to preceding silence (ISI). (**C**) Summary of probability of burst firing relative to proceeding silence (ISI). (**D**) Mean preceding silence as a function of event length. One-way ANOVA, p<0.0001; Tukey's multiple comparisons test, single vs 2 (p<0.0001), 3 (p=0.0001), ≥ 4 (p=0.0004) (**E**) Mean proceeding silence as a function of event length. One-way ANOVA, p=0.0131; Tukey's multiple comparisons test, single vs 2 (p=0.0573), 3 (p=0.0437), ≥ 4 (p=0.2885) SD is represented in the graphs as dotted lines (**B, C**) and the error bars (**D, E**).

The online version of this article includes the following source data for figure 4:

**Source data 1.** Prolonged silent periods precede burst firin.

## Prolonged silent periods precede burst firing

Given that the RB firing reflects a specific, low-activity state of CRH$_{PVN}$ neurons, a question emerged: How do CRH$_{PVN}$ neurons fire in these distinct, short burst trains? To address this, we examined the temporal properties of the burst spike trains by adopting the analysis used by *Harris et al., 2001*. *Figure 4A* illustrates the relationship between ISIs preceding and proceeding individual spikes of the same representative unit shown in *Figure 2A and B*. A cluster of spikes in the lower-left correspond with the spikes in the middle of bursts (i.e. both preceding and proceeding ISI are short). On the other hand, the lower-right cluster represents the initial spike of bursts (long preceding ISI and short proceeding ISI). Notably, the lack of spikes between these two clusters along the X-axis indicate that bursts preferentially start after a long silence period (i.e. preceding ISI >500 ms). By contrast, spikes with short preceding ISI on the left spread along the Y-axis, indicating that the end of bursts are followed by proceeding ISI of variable durations (i.e. not followed by an abrupt start of postburst silence). To express these features more explicitly, we plotted the probability of burst-range firing (*i.e.* proceeding ISI <6 ms) against the preceding ISI for individual units, and then averaged for all bursting units (*Figure 4B*). The graph shows that a burst train seldom initiates without a preceding silent period (*i.e.* ISI) of at least 200 ms, and that the likelihood of burst firing substantially increases when preceded by a silence period longer than 500 ms. In *Figure 4C*, which plots the probability of burst-range firing (*i.e.* preceding ISI <20ms) against proceeding ISI, there was a higher probability of bursting with longer proceeding silences, but the relationship was less prominent than the 'preceding ISI analysis': this likely reflects that burst trains end with gradual prolongation of ISI (i.e. frequency adaptation). We also found that burst firing, regardless of its length, was accompanied by longer preceding silences than SS (0.33 ± 0.16 s, 0.80 ± 0.31 s, 0.89 ± 0.32 s, 1.04 ± 0.43 s, for burst lengths of single, 2, 3, and 4+, respectively; One-way ANOVA, p<0.0001; Tukey's multiple comparisons test, p<0.0001, p=0.0001, p=0.0004 for single vs 2, 3, 4+, respectively. *Figure 4D*). On the other hand, we did not observe these predictive trends in the 'silence' following SS or bursts. Furthermore, the bursts tended to have longer proceeding silences (0.59 ± 0.36 s, 0.57 ± 0.33 s, 0.51 ± 0.40 s, for burst lengths of 2, 3, and 4+, respectively) than SS (0.37 ± 0.18 s; One-way ANOVA, p=0.0131; Tukey's multiple comparisons test, p=0.0573, *P*=0.0437, p=0.2885 for single vs 2, 3, 4+, respectively. *Figure 4E*), which resulted in significant difference only between SS and bursts with three spikes. Again, these data reflect the fact that bursts end with gradual prolongation of ISI and are not followed by an abrupt start of a silent period. These characteristics of burst generation suggest that a prolonged silent period is followed by a brief high-frequency spike trains, which in turn are followed by a prolonged silent period, leading to RB.

## Recurrent inhibitory circuits underlie burst firing

What are the mechanisms generating RB in CRH$_{PVN}$ neurons? Burst firing is known to rely on the intricate interactions between intrinsic neuronal properties, synaptic inputs and network characteristics (*Krahe and Gabbiani, 2004*). Indeed, when studied ex vivo in slice recordings, in which network inputs are substantially lost, CRH$_{PVN}$ neurons do not show overt bursting properties (*Bittar et al., 2019*; *Jiang et al., 2019*; *Khan et al., 2011*; *Luther et al., 2002*; *Matovic et al., 2020*; *Sarkar et al., 2011*; *Wamsteeker Cusulin et al., 2013*). In the present study, we confirmed this lack of ex vivo burst firing phenotype using whole-cell patch-clamp recordings in acute slices. As shown in *Figure 5A* (red traces), in response to depolarizing (monotonic) current steps, CRH$_{PVN}$ neurons fired with a gradual increase in frequency with a moderate adaptation and they did not generate in vivo-like high-frequency bursts. When considering more complex network inputs that dynamically interact with the intrinsic properties of CRH$_{PVN}$ neurons and enable RB firing, our in vivo burst firing pattern predicts the following: (1) the intrinsic properties of CRH$_{PVN}$ neurons favor high-frequency burst firing after a prolonged silence period, and (2) network inputs that cause the prolonged silence contribute to the slow 'rhythms'' of bursting. Taken together, these features are indicative of an inhibitory-driven recurrent network burst mechanism (*McCormick and Feeser, 1990*; *Sherman, 2001*; *Steriade et al., 1993*). Supporting this idea, early in vivo recordings of PVN neurons predicted the existence of recurrent inhibitory circuits based on inhibition following antidromically elicited action potentials (*Saphier and Feldman, 1985*). More recent studies identified a group of local GABAergic interneurons that form recurrent inhibitory circuits (*Jiang et al., 2018*; *Jiang et al., 2019*; *Ramot et al., 2017*). Thus, we hypothesized that recurrent inhibitory circuits underlie the RB of CRH$_{PVN}$ neurons. To test this, we computationally studied the dynamics of single CRH$_{PVN}$ neurons within a network by using a spiking

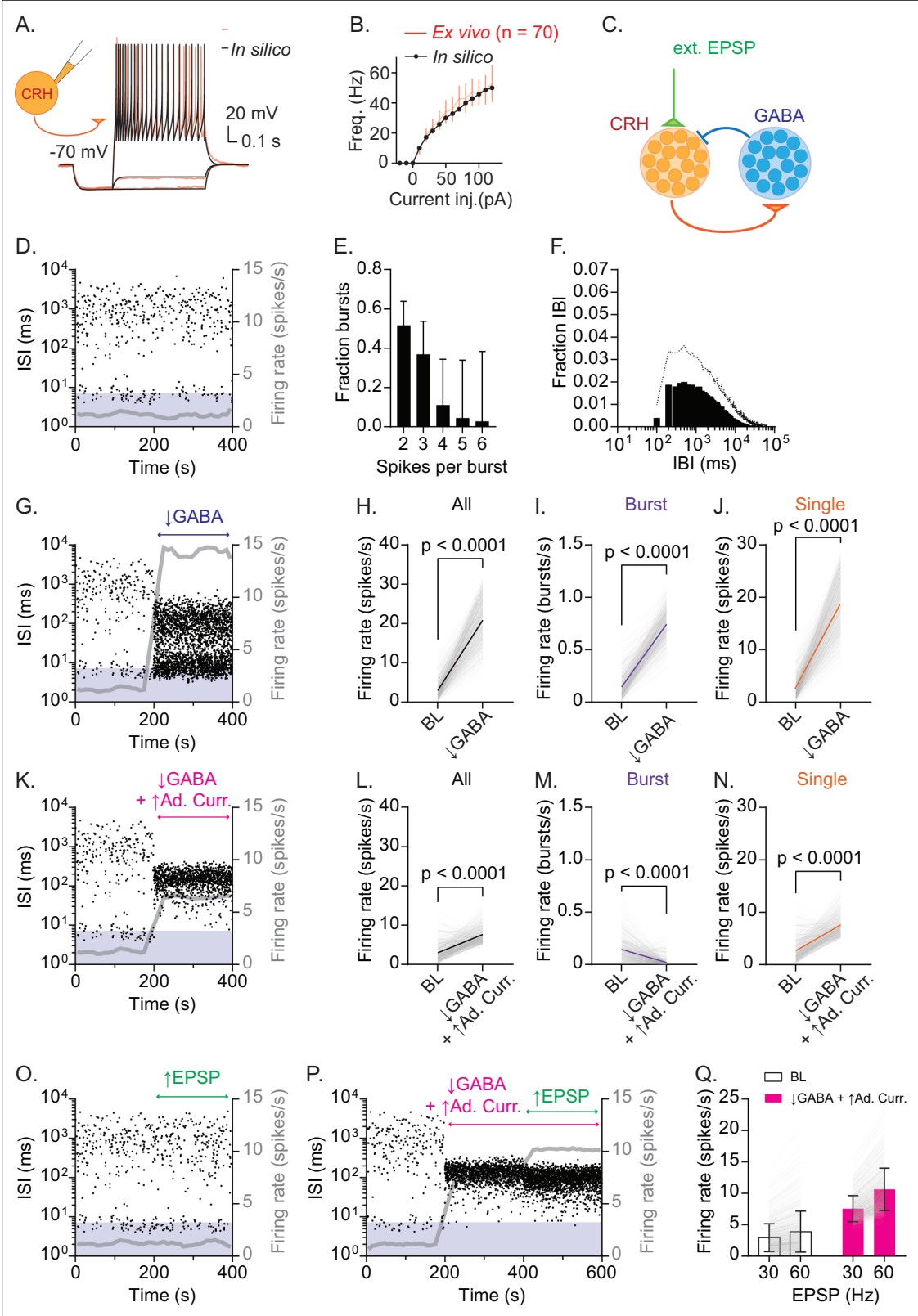

**Figure 5.** Recurrent inhibitory circuits generate burst firing and gate firing response to excitatory inputs in CRH_PVN neurons. (**A**) CRH model neuron in silico (black) fitted to current-clamp recordings of CRH_PVN neurons ex vivo (red). (**B**) F-I curves for in silico (black) and ex vivo (red) neurons. (**C**) Network model diagram. (**D**) Time course of interspike interval (ISI) for a representative model CRH neuron. Gray line is the running average firing rate. (**E**) Summary burst length distribution for model simulations (n=283 for burst rate >0.1 Hz). (**F**) Summary IBI distribution for model simulations (n=283 for

*Figure 5 continued on next page*

*Figure 5 continued*

burst rate >0.1 Hz). (**G**) An example ISI time course before and after a drop of GABA release (Pr 0.8 → 0.1) from 200 s. Gray line is the running average firing rate. (**H–J**) Summary changes in all spikes (H, paired *t*-test, p<0.0001, n=500), bursts (I, paired *t*-test, p<0.0001, n=500) and single spikes (SS) (J, paired *t*-test, p<0.0001, n=500) before (BL) and after the GABA release drop. (**K**) An example ISI time course before and after a drop of GABA release (Pr 0.8 → 0.1) combined with an increase in spike-triggered adaptation current from 200 s. Gray line is the running average firing rate. (**L–N**) Summary changes in all spikes (L, paired *t*-test, p<0.0001, n=500), bursts (M, paired *t*-test, p<0.0001, n=500) and SS (N, paired *t*-test, p<0.0001, n=500) before (BL) and after combined removal of GABA release and increased spike-triggered adaptation current. (**O**) An example ISI time course before and after EPSP frequency increase (30 Hz → 60 Hz) from 200 s. Gray line is the running average firing rate. (**P**) An example ISI time course before and after the combined change in GABA release and spike-triggered adaptation current, followed by an increase in EPSP frequency (30 Hz → 60 Hz) from 400 s. (**Q**) Summary graph for changes in firing rate before (30 Hz) and after EPSP frequency increase (60 Hz) with (white) and without recurrent inhibition (pink). Two-way ANOVA (EPSP × Recurrent inhibition interaction, p<0.0001; Tukey's multiple comparisons test, BL-30 Hz 2.942 ± 2.220 Hz vs BL-60 Hz 3.895 ± 3.252 Hz, p<0.0001; ↓GABA+↑Ad. Curr.-30 Hz 7.548 ± 2.068 Hz vs ↓GABA+↑Ad. Curr.-60 Hz 10.630 ± 3.387 Hz, p<0.0001). SD is represented in the graphs as the error bars.

The online version of this article includes the following source data and figure supplement(s) for figure 5:

**Source data 1.** Recurrent inhibitory circuits generate burst firing and gate firing response to excitatory inputs in CRHPVN neurons.

**Figure supplement 1.** The temporal relationship between network and intrinsic properties with burst firing.

network of adaptive-exponential (AdEx) neuron models (***Brette and Gerstner, 2005***; ***Gerstner and Naud, 2009***; ***Izhikevich, 2003***; ***Markram et al., 2015***). First, the single neuron AdEx models (***Brette and Gerstner, 2005***) were fit to slice patch-clamp recordings of $CRH_{PVN}$ neurons (***Figure 5A*** and ***Table 1***). We evaluated the performance of the model by quantifying the mean squared difference in the subthreshold traces (***Figure 5A***) and the frequency-current (F-I) curve (***Figure 5B***) between the model and experimental data (***Brette and Gerstner, 2005***). Our simulations show that the single neuron models can capture a rapid spike adaptation that follows the first few spikes, timing of repetitive spike firing and the F-I relationship for a series of square pulse depolarizing current injections (***Figure 5A and B***).

Next, the fitted single-neuron models were integrated into a spiking network model to test whether the CRH-GABA neuron network generates rhythmic bursting. We have constructed a network model of the PVN that contains 500 CRH neurons and 500 GABA neurons (***Figure 5C***). The excitatory CRH neurons send a sparse projection to the inhibitory GABA population (2% connection probability), and the GABA population projects back to the CRH neurons with a similar, sparse projection (2%) (***Destexhe, 2009***). CRH neurons also receive noisy random synaptic inputs (30 Hz fast EPSPs), approximated by an instantaneous rise and exponential decay (***Rothman and Silver, 2014***). GABAergic neurons, in turn, receive excitatory CRHergic inputs that have slow rise and decay time constant

**Table 1.** Parameters for spiking network model.

| Neuron parameters | Mean | S | Synapse parameters | Value |
|---|---|---|---|---|
| Number of neurons | 1000 | | $E_e$ (mV) | 0 |
| Number of CRH neurons | 500 | | $E_i$ (mV) | –80 |
| Number of GABA neurons | 500 | | $\tau_e$ (ms) | 12.5 |
| Capacitance (pF) | 22.0 | 2.6 | $\tau_{CRH}$ (ms) | 232.6 |
| $\tau_m$ (ms) | 26.0 | 2.5 | $\tau_i$ (ms) | 20.4 |
| $g_L$ (nS) | 0.9 | 0.19 | $w_e$ (nS) | 3.9 |
| $\Delta_T$ (mV) | 12.1 | 2.0 | $w_{CRH}$ (nS) | 0.005 |
| $E_L$ (mV) | –67.9 | 2.9 | $w_i$ (nS) | 3.3 |
| $V_T$ (mV) | –47.2 | 6.5 | $\tau_{br}$ (s) | 40 |
| $V_R$ (mV) | –58.8 | 2.5 | $\tau_p$ (s) | 80 |
| $\tau_w$ (ms) | 98.2 | 54.3 | | |
| $a$ (nS) | 0.082 | 0.13 | | |
| $b$ (pA) | 17.9 | 9.8 | | |

(~200 ms), modeled as single time constant alpha synapse (*Destexhe et al., 1994*), reflecting the slow CRHR1 mediated excitation (*Jiang et al., 2018*; *Ramot et al., 2017*). The putative network construction was then optimized to fit our in vivo observations. In brief, from a representative unit (shown in *Figure 3K*), temporal subsections were isolated as representative RB and SS firing patterns. Then, in the network model, several rounds of a genetic algorithm were used to optimize the synaptic network parameters; simulations were evaluated using the earth movers' distance (EMD) between the log ISI distributions of the representative spike train and simulated spike trains (Materials and Methods). In this optimized network setting, we found that model CRH neurons generated RB (*Figure 5D*). In the model CRH neurons, the recurrent inhibitory inputs were critical for the long, mostly silent IBI, and the loss of inhibitory inputs (i.e. disinhibition) underlay the timing when excitatory inputs generated depolarization sufficient to trigger burst firing (*Figure 5—figure supplement 1*). In addition to the inhibitory and excitatory synaptic inputs, the spike-triggered adaptation current (*Izhikevich, 2003*) prevented the action potential firing of CRH neurons after the burst firing. However, its influence was less important. This was because its time constant ($\tau_w$=98.2 ms, *Table 1*) was substantially shorter than the time constant of CRHergic excitation ($\tau_{CRH}$ = 232.6 ms, *Table 1*) and ensuing prolonged feedback inhibition. The model CRH neurons, which have heterogeneity in their intrinsic properties obtained in ex vivo recordings (*Table 1*), showed heterogeneity in their burst rates (0.1456 ± 0.1072 bursts/s, n=500). By further characterizing burst firing properties of these bursting model CRH neurons, we found that our network model recapitulated several in vivo features, including burst length and IBI (*Figure 5E and F*). Consistent with these features, RB in the model also constrained the overall firing rate at low levels (~2 Hz, *Figure 5D* gray line). These data show that CRH$_{PVN}$ neurons (which are not intrinsically bursting) fire in RB within a recurrent inhibitory network.

Next, to test the causal roles of recurrent inhibition in constraining firing rate, we performed a series of tests with the network. First, a reduction of recurrent inhibition (by stochastically lowering the release probability [Pr] of GABA→CRH synapses from [0.9–1.0] to [0.01–0.2]) alone with no additional change in external excitatory inputs, robustly increased spiking, and consequently the overall firing rate (*Figure 5G–J*). This indicates that recurrent inhibition constrains the overall firing activity of CRH neurons. However, the removal of recurrent inhibition alone did not eliminate the burst-range high-frequency firing in CRH neurons (*Figure 5G and L*). Interestingly, we found that the in vivo-like switch from RB to SS can be achieved by an additional parameter change, an increase in the 'spike-triggered adaptation current' from (5–18) pA to (36–50) pA (*Brette and Gerstner, 2005*; *Figure 5K–N*). This result predicts potential intrinsic single-cell properties underlying the generation of RB (or its prevention). For example, the spike-triggered adaptation current can represent after-hyperpolarization, (a loss of) after-depolarization, or both (*Izhikevich, 2003*). Next, our modeling data also indicate that recurrent inhibition constrains the response (spike outputs) of CRH neurons to excitatory inputs. To test this idea directly, we increased external synaptic inputs under RB (i.e. in the presence of recurrent inhibition) and SS (i.e. in the absence of recurrent inhibition plus an increase in spike-triggered adaptation current) states (*Figure 6O and P*). In the presence of recurrent inhibition, an increase in excitatory synaptic inputs (from 30 Hz to 60 Hz) caused small change in the overall firing rate, namely CRH neurons' response (*Figure 5O and Q*). By contrast, in the absence of recurrent inhibition, an identical increase in excitatory synaptic inputs robustly increased the firing rate of CRH neurons (*Figure 5P and Q*, Two-way repeated ANOVA, EPSP × recurrent inhibition, interaction p<0.0001), showing that recurrent inhibition works as a gain regulator at CRH neurons.

## Recurrent inhibition generates burst firing in single CRH$_{PVN}$ neurons

Our model simulation predicted that recurrent inhibition is sufficient and necessary for RB and constrains the firing rate of CRH$_{PVN}$ neurons. However, it remains unknown whether our single neuron model, while grounded in direct physiological measurements (*Figure 5AB*, *Table 1*), captures essential biophysical properties required for in vivo-like firing patterns. To directly test this, we went back to ex vivo slice patch clamp electrophysiology and asked if an injection of whole-cell (somatic) currents that mimic the recurrent network inputs of the spiking network model is sufficient to drive RB firing in biological CRH$_{PVN}$ neurons (*Figure 6A*). To this end, we first selected a model CRH$_{PVN}$ neuron that showed representative RB firing (*Figure 6B*), and then extracted the 'network currents' that this model neuron receives within the network model (see Materials and Methods). We then injected these 'network currents' into biological CRH$_{PVN}$ neurons in slices using current-clamp electrophysiology

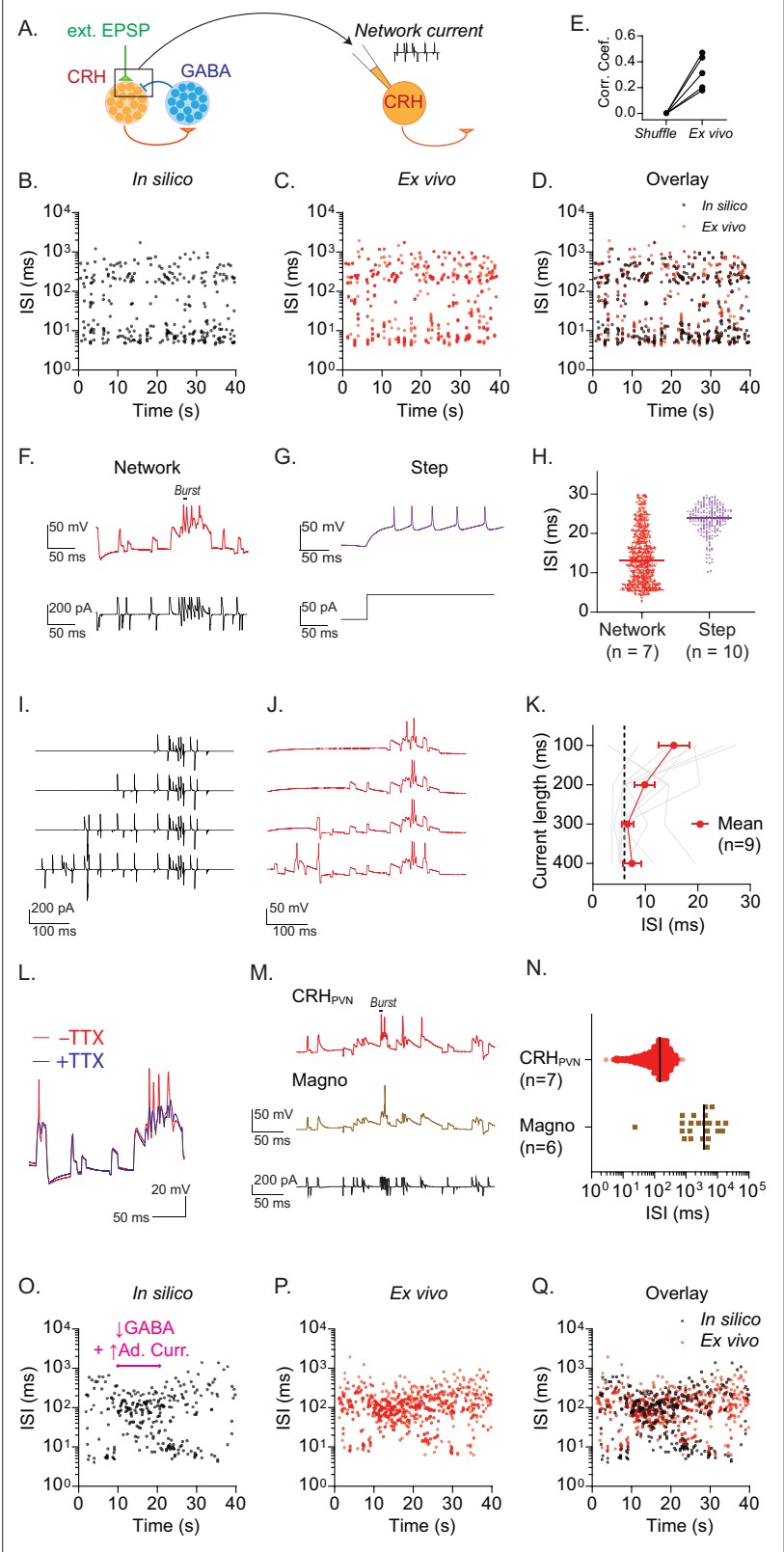

**Figure 6.** Recurrent inhibitory circuits underlie burst firing. (**A**) Diagram of 'network clamp' experiment. (**B–D**) Time course of interspike interval (ISI) for a representative model CRH neurons (**B**), a biological CRH_PVN neuron injected with network current (**C**) and overlay (**D**). (**E**) Correlation coefficient between the time series of model neuron spike train in silico, a biological CRH_PVN neurons' spike train ex vivo, and a shuffled spike train consisting of the model

*Figure 6 continued*

spike train with ISI's shuffled (n=5). (**F, G**) Spike firing patterns in response to network current (**F**) and depolarizing current step (**G**) in a biological CRH$_{PVN}$ neuron. (**H**) ISI distribution of spikes generated by network current (red) and steps of current (purple). Note that spikes with ISI smaller than 30 ms were used for comparison. (**I, J**) Network currents of variable length (**I**) and corresponding membrane voltage changes in CRH$_{PVN}$ neurons (**J**). (**K**) Summary graph for the shortest ISI triggered by network currents of variable lengths. SD is represented in the graph as the error bars. (**L**) CRH$_{PVN}$ neuron's firing response to network current with (blue) and without tetrodotoxin (TTX, 1 μM, red). (**M**) Firing of CRH$_{PVN}$ neuron (orange) and magnocellular neuron (brown) in response to an identical network current (black). (**N**) Summary of ISI triggered by network current in CRH$_{PVN}$ and magnocellular neurons. (**O–Q**) A representative ISI time course for the transition between rhythmic burst and single spiking in a model CRH neuron (**O**), a biological CRH$_{PVN}$ neuron injected with the network current of the model neuron shown in O, and overlay (**Q**).

The online version of this article includes the following source data for figure 6:

**Source data 1.** Recurrent inhibitory circuits underlie burst firing.

---

(*Figure 6C*). We found a remarkable similarity between the RB firing patterns of a model neuron and a biological CRH$_{PVN}$ neuron that received the 'network currents' (*Figure 6D and E*). Notably, in response to square pulse depolarization, biological CRH$_{PVN}$ neurons in acute slices fire in regular spiking patterns (*Figure 6F and H*) with the lowest ISI > 10 ms. The same neurons, however, readily generated burst range high-frequency firing (ISI < 6 ms) in response to complex network current injections derived from the network model (*Figure 6G and H*). A closer examination of the network inputs indicated that high-frequency excitatory inputs drove burst firing (*Figure 6G*). Importantly, however, the high-frequency inputs alone were not sufficient to elicit burst firing, and preceding network inputs were also important. *Figure 6I–K* show that truncated current injection only containing the high-frequency inputs failed to elicit burst firing, and that preceding noisy network inputs (which caused subthreshold membrane potential fluctuations) were necessary to drive the burst firing. These results corroborate with in vivo data that burst firing almost exclusively occurred after a prolonged (>200 ms) silent period (*Figure 4C*), indicating the importance of dynamic interactions between network synaptic inputs (that generate subthreshold membrane potential changes) and intrinsic properties. The burst firing triggered by network synaptic currents were abolished in the presence of voltage-gated Na$^+$ channel blocker tetrodotoxin (TTX, 1 μM, *Figure 6L*), confirming that they were indeed action potentials. Moreover, we found that in magnocellular (non-CRH) neuroendocrine neurons of the PVN, which are known to have ex vivo intrinsic properties (*Luther et al., 2002*) and in vivo firing patterns (*Leng and MacGregor, 2018*) different from CRH$_{PVN}$ neurons, the identical network current failed to trigger the RB (*Figure 6M and N*). Lastly, *Figure 6O* shows an example of model neurons capturing in vivo-like alternations of RB and SS using a transient change in the recurrent inhibition and adaptation current (see in vivo data in *Figure 4J*). The injection of the network currents effectively elicited a similar transition of firing pattern in biological CRH$_{PVN}$ neurons in ex vivo (*Figure 6P and Q*).

## Discussion

Here, we report single-unit activities of identified CRH$_{PVN}$ neurons in vivo for the first time. We show that CRH$_{PVN}$ neurons, within intact circuits, can fire distinct brief, high-frequency (>100 Hz) bursts, a firing pattern that has not been observed in ex vivo slice electrophysiology (*Bittar et al., 2019*; *Jiang et al., 2019*; *Khan et al., 2011*; *Luther et al., 2002*; *Matovic et al., 2020*; *Sarkar et al., 2011*; *Wamsteeker Cusulin et al., 2013*) (but see *Yuan et al., 2019* for slower [<100 Hz] bursts). Guided by the in vivo firing patterns, we have developed a computational model and showed that the RB firing mode reflects critical roles of recurrent inhibition in constraining the overall firing rate of CRH neurons. More generally, the recurrent inhibition controls the gain of CRH neurons' response to excitatory inputs. In biological CRH$_{PVN}$ neurons ex vivo, the injection of whole-cell currents derived from our network model effectively triggered the in vivo-like RB and recapitulated the transition from RB to SS, thus providing direct evidence that intrinsic properties of CRH$_{PVN}$ neurons, which do not overtly fire in burst in slices, are capable of firing in vivo-like RB by interacting with complex inputs reflecting network characteristics. In summary, using a combination of experimental and computational approaches, we demonstrate a novel circuit mechanism that controls state-dependent activity switch of CRH$_{PVN}$ neurons between the baseline and stress.

## CRH<sub>PVN</sub> neurons fire brief bursts with long interburst intervals

Pioneering studies in 1980s used antidromic activation of the median eminence and reported single-unit recordings from parvocellular neuroendocrine neurons of the PVN, a class of neurons that include CRH and several other hormone-releasing neurons, in anesthetized rats (*Day et al., 1985*; *Hamamura et al., 1986*; *Kannan et al., 1987*; *Saphier, 1989*; *Saphier and Feldman, 1985*). Here, using the optrode technique (*Lima et al., 2009*), we report firing activities of identified CRH$_{PVN}$ neurons in anesthetized mice. Overall, our results from CRH$_{PVN}$ neurons agree with the early results from parvocellular neurons in that both populations show spontaneous firing at low rates, and that majority of them increased firing rates in response to stress-mimicking peripheral nerve stimulation.

A key new finding of our study is that CRH$_{PVN}$ neurons fire distinct bursts characterized by RB, a brief train of high-frequency action potentials (~3 action potentials/bursts at >100 Hz) intervened between long periods of silence (constraining the overall firing rate), in addition to SS with more variable and lower spike-to-spike frequencies. This RB firing was not explicitly documented by the earlier studies in rats (*Day et al., 1985*; *Hamamura et al., 1986*; *Kannan et al., 1987*; *Saphier, 1989*; *Saphier and Feldman, 1985*; *Watanabe et al., 2004*); likely because these studies analyzed the firing activities only as time-binned firing rates and did not examine spike-to-spike temporal patterns. In fact, representative traces in one of aforementioned studies showed spikes >100 Hz (see *Figure 4* of Saphier and Feldman, 1985). It should be emphasized that the RB we report here is distinct from 'phasic (burst) firing' which is widely known as a signature firing patterns of vasopressinergic magnocellular neurons of the PVN and SON (*Brown and Bourque, 2006*). The magnocellular phasic burst consists of lower intraburst frequencies (2.5–20 Hz) and lasts much longer with continuous firing (>10 s) than the RB (<100 ms) we report here in CRH$_{PVN}$ neurons (*Poulain and Wakerley, 1982*; *Watanabe et al., 2004*). Also, the phasic burst firing is followed by a prolonged (~10 s) silent period generated by intrinsic mechanisms (*Brown and Bourque, 2006*). We did not observe this type of phasic bursts in identified CRH$_{PVN}$ neurons consistent with the fact magnocellular neurons do not express CRH (*Biag et al., 2012*; *Simmons and Swanson, 2009*).

The brief bursts were intervened with mostly silent IBIs of its median length around 2 s during the RB state. Consequently, despite the high frequency (>100 Hz) burst firing, the time-averaged firing rate of the RB state remained low (3–5 Hz). Furthermore, during non-stress conditions, the RB state dynamically shifted to a non-bursting, SS state where CRH$_{PVN}$ neurons increased their firing rate (10–20 Hz) due to continuous single spiking. A similar shift from RB to continuous SS was evident when CRH$_{PVN}$ neurons increase their firing rate in response to stress-mimicking sciatic nerve stimulations. When considering hormonal CRH levels, the drive for the hormone release is likely to be coded by firing rate over the time scale of seconds and longer, rather than millisecond precisions of firing patterns. Thus, we speculate that persistent SS firing and ensuing elevation of spike rate play a major role in the hormonal CRH release.

## A computational model predicts a novel disinhibition mechanism of CRH<sub>PVN</sub> neurons

Decades of research has established that, under no-stress conditions, the excitability of CRH$_{PVN}$ neurons is tonically constrained by powerful GABAergic synaptic inhibition, and that a release from this tonic inhibition, or disinhibition, is one dominant mechanism for the activation of these neurons, and consequently the HPA axis (*Bains et al., 2015*; *Cole and Sawchenko, 2002*; *Cullinan et al., 2008*; *Inoue and Bains, 2014*; *Levy and Tasker, 2012*; *Roland and Sawchenko, 1993*; *Ulrich-Lai and Herman, 2009*). Multiple disinhibitory mechanisms have been identified, likely reflecting diverse and overlapping neural and hormonal pathways controlling the activities of the HPA axis (*Joëls and Baram, 2009*; *Ulrich-Lai and Herman, 2009*). These include the inhibition of the presynaptic GABAergic neurons (*Anthony et al., 2014*; *Johnson et al., 2019*), depression of GABAergic synaptic terminals (*Ferri and Ferguson, 2005*; *Han et al., 2002*; *Hewitt and Bains, 2006*; *Khazaeipool et al., 2018*) and a depolarizing shift in the equilibrium potential of GABA$_A$ receptor in the postsynaptic CRH$_{PVN}$ neurons (*Hewitt et al., 2009*; *Sarkar et al., 2011*). However, it remained unknown how GABAergic inhibition controls the firing activities of CRH$_{PVN}$ neurons in vivo. In this study, we developed a simple computational model, which was guided by distinct brief burst firing in vivo, and revealed previously underappreciated contribution of recurrent inhibition to the tonic inhibition of CRH$_{PVN}$ neurons. Specifically, the feedback inhibition effectively prevented continuous firing and created a lasting (~2 s) silent periods in

between brief bursts, and consequently constrained the time-averaged firing rate low (~3 Hz). Strikingly, a decrease of recurrent inhibition, with no additional change in the excitatory inputs, profoundly increased the overall firing rate of CRH$_{PVN}$ neurons. In other words, (a loss of) the recurrent inhibition alone partially recapitulates the disinhibitory mechanisms of CRH$_{PVN}$ neurons. That said, there is more to consider regarding the activity of CRH$_{PVN}$ neurons, as the removal of recurrent inhibition alone results in continuous burst-range high frequency firing, which does not typically occur during stress-induced firing increase in vivo (*Figure 3A–F*). Thus, our model predicts that the in vivo-like transition from RB to SS requires an increase in spike-triggered adaptation current (*Brette and Gerstner, 2005*), pointing to single-neuron intrinsic properties underlying burst generation. For example, the spike-triggered adaptation current can represent after-hyperpolarization, (a loss of) after-depolarization, or both (*Brette and Gerstner, 2005*).

It should be noted that our model for the recurrent excitatory-inhibitory circuit in the PVN is simplified and does not incorporate external inhibitory inputs to CRH$_{PVN}$ neurons or external excitatory and inhibitory inputs to the GABAergic neurons (*Ulrich-Lai and Herman, 2009*). Furthermore, transmission from CRH neurons to GABAergic neurons is exclusively modeled by slow CRHergic transmission based on recent experimental data (*Jiang et al., 2019*; *Ramot et al., 2017*), but CRH$_{PVN}$ neurons have been shown to release glutamate at some synapses (*Füzesi et al., 2016*). Despite these simplification, our model parsimoniously captures the key features of firing behavior in vivo, including the brevity of burst episodes and long IBI (*Figure 5E and F*). Thus, our simplified model, which is the first network model for CRH$_{PVN}$ neurons to our best knowledge, is a useful first step to understand the spiking behaviors of CRH$_{PVN}$ neuron in a network, and to generate experimentally testable hypotheses for future research.

So, what are the identities of the GABAergic neurons that form recurrent connectivity with CRH$_{PVN}$ neurons? Indeed, the recurrent inhibitory circuits for CRH$_{PVN}$ neurons has been first suggested by an early in vivo electrophysiology recordings of parvocellular neurons in rats based on an inhibition following action potentials antidromically activated from the median eminence (*Saphier and Feldman, 1985*). Recent studies have revealed local recurrent circuits within the PVN, where CRH$_{PVN}$ neurons excite CRHR1-expressing GABAergic interneurons, which in turn send GABAergic inputs to CRH$_{PVN}$ neurons (*Jiang et al., 2018*). CRH$_{PVN}$ neurons excite CRHR1-expressing neurons via slow metabotropic actions with little, if any, fast (glutamatergic) excitatory synaptic transmission (*Jiang et al., 2018*; *Ramot et al., 2017*). Notably, our network model found that the long IBI (~2 s) of brief bursts require slow excitatory EPSPs consistent with metabotropic signaling ($\tau_{CRH}$ >200 ms) rather than fast ionotropic EPSPs ($\tau_e$ <20 ms). In addition to this intra-PVN microcircuit, it is possible that CRH$_{PVN}$ neurons form additional recurrent inhibitory circuits. Indeed, there is growing appreciation that CRH$_{PVN}$ neurons send projections within the brain besides their well established projections to the median eminence (for hormonal CRH release) (*Füzesi et al., 2016*; *Jiang et al., 2018*; *Jiang et al., 2019*; *Kim et al., 2019a*; *Li et al., 2020*; *Ono et al., 2020*; *Ramot et al., 2017*; *Rho and Swanson, 1989*; *Yuan et al., 2019*). For example, the perifornical area (*Füzesi et al., 2016*; *Rho and Swanson, 1989*) and the lateral hypothalamus (*Li et al., 2020*; *Ono et al., 2020*) receive direct synaptic inputs from CRH$_{PVN}$ neurons: these brain areas in turn send direct GABAergic inputs to CRH$_{PVN}$ neurons (*Boudaba et al., 1996*; *Cullinan et al., 2008*; *Roland and Sawchenko, 1993*; *Ulrich-Lai and Herman, 2009*).

## Synaptic activity underlies flexible firing patterns of CRH$_{PVN}$ neurons in vivo

In acute slices ex vivo, CRH$_{PVN}$ neurons typically show a SS electrophysiological phenotype, or 'regular' spiking, with up to around 50 Hz of spike-to-spike frequency in response to steps of intracellular current injections in mice and rats (*Bittar et al., 2019*; *Jiang et al., 2019*; *Khan et al., 2011*; *Luther et al., 2002*; *Matovic et al., 2020*; *Sarkar et al., 2011*; *Wamsteeker Cusulin et al., 2013*). Thus, our in vivo recordings of the RB (>100 Hz spike-to-spike frequency) revealed a previously underappreciated repertoire of the firing patterns of CRH$_{PVN}$ neurons. In this regard, CRH$_{PVN}$ neurons are different from thalamic relay neurons that have been extensively studied for their properties to switch between bursting and single (tonic) spiking modes both in vivo and ex vivo (*McCormick and Pape, 1990*; *Sherman, 2001*; *Steriade et al., 1993*). The difference can be partly explained by the fact that CRH$_{PVN}$ neurons express only modest levels of T-type currents (*Luther et al., 2002*) and little, if any, H-currents (see *Figure 5A*): these intrinsic properties play crucial roles in the generation of rhythmic burst firing,

and switch between burst and single spiking modes in thalamic relay neurons (*McCormick and Pape, 1990*; *Sherman, 2001*; *Steriade et al., 1993*).

In other words, our data (*i.e.* the absence of overt burst-generating intrinsic properties) suggest that the intrinsic properties of $CRH_{PVN}$ neurons conduct flexible input/output signal processing by interacting with the physiological network dynamics which is substantially lost in ex vivo slice preparations. Supporting this idea, our modeling showed that model CRH neurons, which is fitted to 'regular' spiking phenotype characterized ex vivo, readily fire in RB when they receive the bombardment of excitatory inputs (30 Hz EPSPs) combined with the recurrent inhibition. Thus, our modeling results demonstrate that as simple as two types of network-like inputs are sufficient to generate a firing pattern that is qualitatively different from and not readily evident in conventional (*e.g.* square pulse) electrophysiological characterizations ex vivo.

Interestingly, Yuan et al. recently reported spontaneous 'burst firing' of $CRH_{PVN}$ neurons ex vivo using cell-attached recordings in slices prepared from repeatedly stressed mice (*Yuan et al., 2019*). It should be noted, however, that the intraburst frequency of this ex vivo 'burst firing' was <100 Hz (averaging around 50 Hz), which was different from the in vivo RB (>100 Hz intraburst frequency) we report here in non-stressed mice. Furthermore, the ex vivo 'burst firing' involved stress-induced plasticity that developed over days because it only became evident after three daily foot shock, but was absent when mice were not previously stressed, or even after two daily foot shock. Accumulating evidence show that $CRH_{PVN}$ neurons undergoes diverse forms of synaptic and intrinsic plasticity in response to acute as well as during chronic stress (*Bains et al., 2015*; *Herman and Tasker, 2016*; *Matovic et al., 2020*; *Salter et al., 2018*; *Yuan et al., 2019*). Thus, it is an important direction of future studies to examine the plasticity of $CRH_{PVN}$ neurons' firing in vivo and their contributions to (mal)adaptations of the stress response.

Our computational modeling addressed the potential interplay between in vivo-like network inputs and the intrinsic properties of $CRH_{PVN}$ neurons. The flip side of our approach, however, is the over-simplification of the network inputs and questions about their physiological relevance. To directly address the latter concern, we applied the modeling results to biological neurons ex vivo. Specifically, we extracted network inputs from representative model neurons that fire RB, and then injected the network current waveform into biological neurons under whole-cell current-clamp configuration ex vivo. We term this new approach as 'network clamp'. Network clamp injects a predetermined current waveform which can be readily available from the leaky-integrate-and-fire model, and therefore much simpler, operationally, and computationally, than dynamic clamp. One limitation is that the amplitude of the current fluctuations can be larger than physiologically realistic range. Despite this limitation, however, we found that the network current generated RB in $CRH_{PVN}$ neurons and the transition to SS upon changes in network parameters (*i.e.* removal of recurrent inhibition and increase in adaptation current), providing direct evidence for the capability of $CRH_{PVN}$ neurons to fire with high frequency bursts (*i.e.* RB pattern) and also bidirectionally switch to regular spiking (*i.e.* SS pattern). Importantly, the burst firing does not simply occur as a results of current injection patterns but requires unique interaction between the intrinsic properties of $CRH_{PVN}$ neurons and the network inputs. This is supported by the following two controls. First, network inputs, which drove subthreshold membrane potential changes preceding (>200 ms) the burst firing, were necessary because truncated (<200 ms) network currents, which only contain high-frequency synaptic inputs component, failed to elicit the brief bursts in $CRH_{PVN}$ neurons (Fig. 6I-K). Notably, this result corroborates well with in vivo data that RB almost exclusively occur after long (>200 ms) silent period (*Figure 4B*). Second, magnocellular neurons, a separate class of neuroendocrine neurons of the PVN with different intrinsic properties (*Luther et al., 2002*), did not fire in brief bursts in response to the same network current injection that elicited brief bursts in $CRH_{PVN}$ neurons (*Figure 6M and N*). The intrinsic properties key to this burst firing are currently unknown and warrants future investigation.

These results point to future applications of the 'network clamp' in order to examine specific biophysical mechanisms that interact with network inputs to generate in vivo-like firing patterns in other brain areas. It is well known that neurons in slice differ in their response properties from intact in vivo preparations (*Destexhe and Paré, 1999*; *Destexhe et al., 2003*) and that injection of noise approximating the inputs from the large number of synaptic contacts in intact preparations can recreate firing conditions observed in vivo (*Destexhe and Rudolph-Lilith, 2012*; *Destexhe et al., 2003*; *Rudolph and Destexhe, 2003*; *Rudolph and Destexhe, 2006*). Advanced techniques have been developed

to measure neuronal firing rate responses under injection of theoretically motivated noise processes (*Zerlaut et al., 2016*). In this work, however, we have focused on injecting the time-varying inputs from a *single* cell in the network model to a biological neuron, so that we can directly compare the activity patterns evoked in the biological and simulated cells. We find that computationally generated inputs can recreate the specific burst firing mode we have reported in this work. In this way, this injection protocol bears some similarity to the 'iteratively constructed network (ICN)' developed by *Reyes, 2003* to provide a direct test of whether biological cells can propagate synchrony without some of the caveats that can be imposed by computational simulations (*Rudolph and Destexhe, 2007*). We suggest that this 'network clamp' protocol has more general applicability; however, for validating spiking network models of neural circuits. In this work, the spiking network model provided a testable prediction: initially, biological neurons recorded ex vivo did not exhibit bursting behavior when stimulated with step current pulses, but would they burst when driven with network-generated input? In this case, the biological neuron in this 'network clamp' paradigm exhibited behavior consistent with this prediction, strengthening confidence that our spiking network model captures a meaningful underlying mechanism in this system. Because this paradigm allows to test multiple model predictions in a single setup (for conventional patch-clamp electrophysiology), and because it also opens the possibilities for testing how neuronal firing and response patterns may change with neuromodulation applied in the slice, we suggest that the model-experiment protocol developed and tested in this work may be able to distinguish between multiple models that can account for a single dataset in isolation (*Marder and Goaillard, 2006*).

## Limitations of the study

Our in vivo recordings were performed under urethane anesthesia, as was the case for the majority of previous studies using in vivo recordings from the PVN in anesthetized rats (*Hamamura et al., 1986*; *Kannan et al., 1987*; *Saphier, 1989*; *Saphier and Feldman, 1985*; *Saphier and Feldman, 1988*; *Saphier and Feldman, 1990*). Urethane produces a long-lasting steady level of surgical anesthesia, and preserves the subcortical and peripheral neural functions (*Maggi and Meli, 1986*), making it suitable for long electrophysiological recordings from the hypothalamus. For example, under urethane anesthesia in rabbits, hypothalamic neurons have been shown to readily respond to varieties of stress stimuli including hypoxia, hypercapnia, loud noise, and pain (*Cross and Silver, 1963*). Of particular relevance to our study, sciatic nerve stimulation (under urethane anesthesia) has been shown to effectively elevate blood ACTH levels (*Hamamura et al., 1986*), validating that the stress-responsiveness of the HPA axis is preserved. However, it should be noted that urethane has also been shown to enhance autonomic (*Shimokawa et al., 1998a*) and HPA axis activity (*Hamstra et al., 1984*), likely by enhancing adrenergic inputs to the PVN (*Shimokawa et al., 1998b*). Considering that (nor)adrenaline has both excitatory and inhibitory effects on PVN neurons (*Han et al., 2002*; *Itoi et al., 1994*; *Saphier and Feldman, 1991*) and that it can potently influence state-switch between bursting and single spiking (*Pape and McCormick, 1989*), urethane may have affected the baseline firing activities and burst firing properties of CRH$_{PVN}$ neurons in general. Future studies using awake animal recordings are warranted.

## Ideas and speculations

Our study found brief, high-frequency burst firing of CRH$_{PVN}$ neurons in vivo. This unexpected finding led us to propose, using a tight combination of experimental and computational approaches, that recurrent inhibition can generate the brief bursts and constrain prolonged firing activities of CRH$_{PVN}$ neurons leading to hormonal release. Then, what is the advantage of a RB mechanism during periods when CRH neuron activity is not required for the hormone release? Does RB firing encode and convey specific information? One possibility is that burst firing facilitates the self-maintenance of the RB firing state. That is, a brief burst enables high-fidelity synaptic transmission (*Lisman, 1997*) and efficiently excites GABAergic neurons that return recurrent inhibition to CRH$_{PVN}$ neurons: this feedback inhibition, in turn, facilitates subsequent burst firing. Notably, our spiking network model found that recurrent inhibition works as a gain regulator at CRH neurons (*Figure 5*). Thus, our model prediction offers a new circuit mechanism for the experimental data that disinhibition, more so than increasing excitatory inputs, is crucial for the activation of CRH$_{PVN}$ neurons for hormonal release (*Cole and Sawchenko, 2002*; *Hewitt et al., 2009*; *Kovács et al., 2004*; *Sarkar et al., 2011*). Beyond constraining hormonal

release of CRH, the idea that CRH$_{PVN}$ neurons use brief bursts for high-fidelity synaptic transmission, without triggering massive HPA axis activation, raises an intriguing possibility in light of emerging new roles of CRH$_{PVN}$ neurons that are independent of hormone release. Recent studies showed that CRH$_{PVN}$ neurons are involved in controlling wakefulness (*Ono et al., 2020*), valence encoding (*Kim et al., 2019a*), reward processing (*Kim et al., 2019a*; *Yuan et al., 2019*), and defensive behavior control (*Daviu et al., 2020*; *Füzesi et al., 2016*) under non-stress and stress conditions. We propose that brief intermittent bursts of activity are ideally suited to drive rapid behavioral and emotional changes independently of systemic and long-lasting hormonal response.

## Materials and methods

### Animals

All experimental procedures were performed in accordance with the Canadian Council on Animal Care guidelines and approved by the University of Western Ontario Animal Use Subcommittee (AUP: 2018–130). Homozygous *Crh-IRES-Cre (B6(Cg)-Crhtm1(cre)Zjh/J)* mice (Stock No: 012704, the Jackson Laboratory) were crossed with homozygous Cre-reporter *Ai14 (B6.Cg-Gt(ROSA)26Sortm14(CAG-TdTomato)Hze/J)* mice (Stock No: 007908, the Jackson Laboratory) to produce CRH-TdTomato reporter offspring. The specificity of cre-expression in the PVN in these mice has been characterized in previous studies (*Chen et al., 2015*; *Wamsteeker Cusulin et al., 2013*). 10 adult male mice (>60 days old) were used for viral injections and recordings. Animals were group-housed (2–4 per cage) in standard shoebox mouse cages with ad libitum access to food and water. Animals were housed on a 12 hr dark/12 hr light cycle (lights on from 07:00 to 19:00) in a temperature-controlled room (23 ± 1°C).

### Viral injection

To express channelrhodopsin-2 (ChR2) in CRH$_{PVN}$ neurons, we used an AAV carrying the EF1α promoter and the double-floxed inverted ChR2(H134R)-EYFP coding sequence that is inverted and turned on by cre-recombination (AAV2/5- EF1α-DIO-ChR2(H134R)-EYFP). The plasmid was a gift from Dr. Karl Deisseroth http://n2t.net/addgene:20298; RRID:Addgene_20298. The AAV preparation was obtained from the Neurophotonics Centre (5 × 10$^{13}$ GC/ml, Laval University, Canada).

For AAV injection, the animal was anesthetized under isoflurane (2%) using a low flow gas anesthesia system, (Kent Scientific Corporation) and placed in a stereotaxic apparatus on a heating pad. A finely pulled glass capillary was loaded with the virus and slowly lowered into the brain of animals, targeting the PVN on each side of the brain (A/P: –0.70 mm, M/L:±0.25 mm, and D/V:–4.75 mm from bregma). A total of ~240 nL (41.4 nL × 6 at 23 nL/s) was pressure-injected on each side using the Nanoject II (Drummond Scientific Company). The pipette was held in place for 5 min after injection to allow diffusion before being slowly retracted from the brain. The incision was closed by suturing and the animals were injected with analgesic (buphrenorphone 0.1 mg/kg, s.c.) at the end of surgery. Animals were allowed to recover for at least 6 weeks for optimal ChR2 expression before electrophysiological recordings.

### Electrophysiology

Animals were initially anesthetized under isoflurane (1–2%) and urethane (1.5 g/kg in 0.9% saline, intraperitoneal) to perform surgery for sciatic nerve isolation and a craniotomy/durotomy for insertion of the recording probe into the brain. We first isolated the sciatic nerve (see details in Sciatic nerve stimulation), and then the probe was lowered into the brain. Thereafter, the animal was taken off isoflurane. The probe was vertically inserted above the PVN (A/P: –0.70 mm from bregma, M/L:±0.25 mm from bregma) and slowly lowered ventrally to the PVN (target D/V: –4.5 mm from cortical surface, photo-tagged units (see details in Optogenetic identification of CRH neurons) were found between –4.20 mm and –4.80 mm from cortical surface).

Extracellular neural signals were recorded using a single shank, 32-channel (8 rows × 4 columns) silicon probe, with recording sites spanning 60 µm in depth (Cambridge NeuroTech). An optic fiber (100 µm core diameter, 0.37 NA, A45; Doric Lenses) was attached by the manufacturer parallel to the electrode shank with a vertical offset of 250 µm, constituting an optrode. The electrode was connected to a digital headstage (ZD ZIF-Clip; Tucker-Davis Technologies) with an internal Intan amplifier chip. The digitized signals were sent to the amplifier (PZ5; Tucker-Davis Technologies), which in

turn connects out to the processing unit (RZ5D, Tucker-Davis Technologies) via a fiber optic cable. Broadband signals were sampled at 25 kHz and bandpass filtered between 300 and 3000 Hz. The threshold for spike detection was a minimum three SDs above the noise floor. The waveforms for detected spikes were sorted offline (Offline Sorter, Plexon) using manual and automatic clustering (T-Distribution E-M). To ensure the quality of single-unit isolation, only discrete clusters with L-ratios <0.05 (in 2D or 3D spaces) was included for single-unit analysis.

## Optogenetic identification of CRH neurons

Transistor-transistor logic (TTL) signals to trigger optogenetics light stimulation were sent from the signal processing and acquisition software Synapse (Tucker-Davis Technologies) to a LED driver (PlexBright LD-1 Single Channel LED Driver, Plexon). The LED driver controlled the intensity of light emitted from the LED module (PlexBright Table-Top LED Module Blue 465 nm, Plexon) which was coupled to an optic patch cable (200 μm core, PlexBright Optical Patch Cable, Plexon). The patch cable connected via a plastic sleeve to the optic fiber of the optrode.

For each electrode location, 5 ms and 50 ms single light pulses were delivered at 2 mW. Following offline sorting of waveforms, ChR2-expressing CRH single units were identified by their time-locked responses to light (see details in Data analyses and statistics: Identification of light-evoked spikes). Specifically, we clustered all waveforms (both spontaneous and light-evoked in a blinded manner) for single unit isolation (L-ratios <0.05). Thereafter, single units clustered with the light-evoked waveforms were considered as light responsive and thus CRH neurons, and other single units simultaneously recorded with light responsive neurons but did not respond to the light were considered as light non-responsive, which includes both non-CRH neurons and CRH neurons with insufficient ChR2 expression and/or light exposure.

## Sciatic nerve stimulation

Under isoflurane anaesthesia, a small incision was made into the hind limb of the animal and the sciatic nerve was isolated from surrounding tissue. A bipolar tungsten stimulating electrode was gently placed on the nerve and the nerve was kept hydrated with periodic applications of sterile saline. The contralateral nerve to the recording site was stimulated with 1.6 mA negative pulses at 20 Hz (5 × 0.5 ms) using a pulse stimulation unit (S88, GRASS Instrument Co) connected to a stimulus isolation unit (Model PSIU6, GRASS Instrument Co) triggered via TTL signals.

## Histology

At the end of recording, animals were euthanized with an overdose of pentobarbital sodium (150 mg/kg, i.p.) and transcardially perfused with 0.9% saline and 4% paraformaldehyde (PFA). The brain was collected and left to postfix in 4% PFA at 4°C overnight. The brain was washed in phosphate buffer saline (PBS) solution and sliced into 40 μm coronal sections and counter stained with 4,6-diamidino-2-phenylindole (DAPI). The slices were mounted and verified for CRH expression (tdTomato), ChR2 expression (eYFP), and the dye-painted electrode tract (Vybrant DiD, Thermo Fisher V22889).

## Experimental design

Each recording started with the baseline recordings of spontaneous firing without any intentional sensory stimulation for at least 10 min. This was followed by light stimulation for CRH neuron identification, then subsequently sciatic nerve stimulation (minimum 3 min between light stimulation and sciatic nerve stimulation).

## Data analyses and statistics

Data analyses were carried out using built-in and custom-built software in MATLAB (MathWorks). Data analysis codes are available on GitHub, copy archived at swh:1:rev:d4d153e44882a0c2a7d0a1e-f11193347aa5d6c96 (*Ichiyama and Mestern, 2022*). Graphs were made and statistical analysis were conducted using Prism 8 (GraphPad).

### Identification of light-evoked spikes

Time-locked response to light stimulation was tested by an increase in spiking activity to a 5 ms pulse of blue light (25 trials). The response was plotted on a peristimulus time histogram (PSTH) aligned to

the onset of light. Peristimulus frequency before and after light onset (20 ms windows) was compared by a trial-by-trial paired *t*-test: a significant increase was defined as light-responsive. The event probability before and after light (20 ms windows) was calculated as the number of trials that had at least 1 event within the response window, divided by the total number of trials.

## Spike pattern analysis

For each single-unit, spike activity was classified into bursts and SS. Bursts were detected as series of two or more spikes starting with the initial ISI less than 6 ms and subsequent ISIs less than 20 ms. Although rare, to prevent contamination of bursts by high-firing SS which gradually speed up (*i.e.* shortening ISIs) into the burst range, we adopted a secondary criterion for bursts to be preceded by an ISI >25 ms. Spikes not associated with bursts were defined as SS. For analyses involving burst rates (*i.e.* burst rate, burst index, IBI), each burst was counted as a single event regardless of burst length (the number of spikes per burst). IBI distribution was first calculated for each cell (100 ms bins, plotted by upper limit of bin), and then the group distribution was generated by plotting the mean and standard deviation. Similarly, the distribution of burst length (the number of spikes per burst) was first calculated for each cell, and then the group distribution was generated by plotting the mean and SD.

To examine spiking dynamics that may influence burst activity, the probability of burst initiation (ISI$_{(i)}$<6 ms) as a function of preceding silence (ISI$_{(i-1)}$) was plotted for each cell. To do this, spikes were binned by ISI$_{(i-1)}$ (10 ms and 500 ms bins for periods <500 ms and ≥500 ms, respectively), and the probability of ISI$_{(i)}$<6 ms was calculated for each bin for each cell. Then, the group average was plotted with the mean and SD. To examine the influence of preceding silence on burst length, spikes were binned by event length, and the mean preceding silence was calculated for each cell. Then, the group average was plotted with the mean and SD. The silent period analysis was repeated for proceeding silent periods. In this case, the probability of burst (ISI$_{(i)}$<6 ms) as a function of the proceeding silence (ISI$_{(i+1)}$) was plotted. Finally, the mean proceeding silence was compared by event lengths.

## Firing rate analysis

The average baseline firing rates were calculated as the total number of spikes divided by total time during the spontaneous baseline recording (10 min). The firing rate time course was calculated as the number of spikes in every 10 s bins. The running average firing rate represents a moving average of five consecutive bins (bin*[i-2]*+bin*[i-1]*+bin*[i]*+bin*[i+1]*+bin*[i+2]*/5) and was used to visualize spontaneous fluctuations of firing rate. These fluctuations in firing rate were then overlaid onto ISIs plotted across time to visualize the relationship between firing rate to firing patterns. To examine the relationship between burst rate and total spike rate, for every 10 s bin in individual units, total spike rate was plotted against burst rates of 0.1, 0.2, 0.3, 0.4, 0.5, or ≥ 0.6 bursts/s. The relationship was plotted for individual unit (***Figure 3L***), as well as pooled across units (***Figure 3M***, n=18).

## Sciatic nerve stimulation

The response to sciatic nerve stimulation was plotted on PSTHs aligned to the onset of stimulation (the first of the 5-pulse train). A paired t-test was performed between the average post-onset firing rates (2 s across 30–60 trials) and the baseline firing for each cell. Responses to sciatic nerve stimulation was analyzed in three categories: all spikes regardless of spiking pattern, SS only, and bursts only (each burst train counted as a single event regardless of the burst length).

## **Recurrent inhibitory network model**

### Single neuron model fit

Single neurons were modeled using the Adaptive Exponential Integrate and Fire (AdEx) model (***Brette and Gerstner, 2005***).

$$C\frac{dv}{dt} = g_L\left(E_L - v\right) + g_L\Delta_T \exp\left(\frac{v - V_{th}}{\Delta_T}\right) + g_e\left(E_e - v\right) + g_i\left(E_i - v\right) - w + I \tag{1}$$

$$\begin{aligned} v &\to v_r \\ w &\to w + b \end{aligned} \tag{2}$$

Here, $C$ is the membrane capacitance, $w$ is an adaptation variable, $I$ is the applied current, $g_L$ is the leak conductance, $E_L$ is the resting membrane potential, $\Delta_T$ is the slope factor, and $V_{th}$ is the threshold potential. The slope factor determines the sharpness of the threshold. The membrane potential $v$ is modeled as a sum of these conductance parameters (*Equation 1*). When the membrane potential $v$ exceeds the threshold $V_{th}$ the neuron emits a spike, and the neuron is reset according to *Equation 2*. $w$ is computed as a function of the subthreshold, and spike-triggered adaptations, respectively (*Equation 3*)

$$\tau_w \frac{dw}{dt} = a\left(v - E_L\right) - w \tag{3}$$

where $\tau_w$ is the time constant and $a$ is subthreshold adaptation conductance.

To refine the model to capture the behavior of CRH$_{PVN}$ neurons, we fit the model with whole-cell patch-clamp recordings of CRH$_{PVN}$ neurons in acute brain slices (n=33, 5 mice) using methods detailed in *Matovic et al., 2020*. Recordings consisted of 14 trials containing a hyperpolarizing (–20 pA for 200 ms) pulse followed by a stepped depolarizing current pulse (10 pA steps for 500 ms). The same stimulus was used in subsequent model simulations. First, the firing rate-current (F-I) curve for the model was simulated across a large parameter space. Then, the posterior distribution for the parameter space was fit using the Sequential Neural Posterior Estimation (SNPE) method as previously described (*Gonçalves et al., 2020*). Second, for each reference cell, the F-I curve was used to draw a restricted parameter space from the previously fit posterior. Then, 200 rounds of evolutionary optimization were used to refine the model (*Lynch and Houghton, 2015*). The penalty (or evaluation of fit) was calculated as the sum of the mean squared error (MSE) between the subthreshold voltage of the model and reference, and the log MSE between the F-I curve of the model and reference. Models that produced an MSE above 5 mV (for the subthreshold voltage) and above 10 Hz (for the F-I curve) were considered poor fits, and were discarded. To produce a robust general model, the average of the parameter fit for all reference traces was used in the network model (*Table 1*).

## Network model

To construct the network model, we utilized a simplified recurrent network template with two distinct populations of AdEx neurons (*Destexhe, 2009*). These neurons consisted of a modified AdEx model with added excitatory and inhibitory synaptic parameters (*Equation 1*).

$$\begin{aligned} g_e &\to g_e + w_e \\ g_i &\to g_i + w_i \end{aligned} \tag{4}$$

$$\begin{aligned} \tau_e \frac{dg_e}{dt} &= -g_e \\ \tau_i \frac{dg_i}{dt} &= -g_i \end{aligned} \tag{5}$$

Here, $g_e$ & $\tau_e$ and $g_i$ & $\tau_i$ represent the inhibitory and excitatory conductance and time constant, respectively. Following a presynaptic spike, the inhibitory and excitatory conductance were incremented according to *Equation 4*, followed by a loss of conductance via a simple exponential decay (*Equation 5*).

Both populations were interconnected, with one population presenting as excitatory and one as inhibitory. For simplicity, only the excitatory neurons received external input. First, we constrained the excitatory portion (n=500) of the network to our biologically realistic CRH$_{PVN}$ neurons. To do so, we used previously fit parameters from the single neuron model as described in Single neuron model fit. Next, the remaining inhibitory population was constrained with parameters, reflecting the tonic spiking GABAergic neurons of the thalamus (*Destexhe, 2009*). Changes in adaptation current $b$ were modeled with instantaneous increment followed by an exponential decay function (*Equation 6*), reflecting the slow change in intracellular parameters.

$$\begin{aligned} \tau_b b_{inc} &= -b_{inc} \\ b &\to b_{intial} + b_{inc} \end{aligned} \tag{6}$$

For each synapse, the probability of release $p$ was modelled by the logistic differential equation

$$\frac{dp}{dt} = \frac{p}{\tau_p}\left(1 - \mathrm{p}\right)$$
$$p \rightarrow 0.1 \tag{7}$$

Based on previous literature, the excitatory input of CRH$_{\mathrm{PVN}}$ neurons to the GABAergic population was modeled to be mediated by the CRHR1 receptor (*Jiang et al., 2018*; *Jiang et al., 2019*; *Ramot et al., 2017*). Therefore, this synapse was modeled by a slow-growing alpha function, a common model of neuromodulator function (*Destexhe et al., 1994*):

$$\frac{d[\mathrm{CRH}]}{dt} = \frac{y - w_{\mathrm{CRH}}}{\tau_{\mathrm{CRH}}}$$
$$\frac{dy}{dt} = \frac{-y}{\tau_{\mathrm{CRH}}}$$
$$y \rightarrow [\mathrm{CRH}] + w_{\mathrm{CRH}} \tag{8}$$

The remaining free parameters $w_e$, $w_i$, $\tau_e$, $\tau_i$, $\tau_{\mathrm{CRH}}$, and the EPSP input frequency were tuned with respect to the observed extracellular spike trains in vivo. To begin with, we selected a representative opto-tagged CRH neuron baseline recording as reference (*Figure 2B*). Next, we selected two characteristic (50 s) sections of the reference spike train that reflected bursting and single spiking independently. Then, several rounds of evolutionary algorithms were used to tune the parameters. For each round, the network was simulated with the given parameters with an increased adaptation current (2 × 50 s). Then, the distance between simulated and reference data was computed as the earth mover's distance (Wasserstein metric) between the log-scaled ISI distributions (*Olkin and Pukelsheim, 1982*). The total distance was taken as the sum of the smallest 100 distances between each simulated unit and the reference. The standard adaptation current and the increased adaptation current simulations were compared to the bursting and tonic references, respectively.

## Generation of network clamp input for ex vivo experiments

To investigate the ability of our model to induce burst in biological neurons ex vivo, we extracted the input currents from a model neuron in the network. First, a model neuron, isolated within the network, had its initial resting membrane potential V$_{m\emptyset}$ and leak reversal E$_L$ constrained to the mean resting membrane potential of the CRH$_{\mathrm{PVN}}$ neurons in our recording conditions ex vivo (−67.9 mV, *Table 1*). Then, the model was simulated for 50 s, and the model neuron was allowed to respond to the incoming input freely. Every time-step (100 μs), the sum of the incoming excitatory, inhibitory, and adaptation currents were extracted following *Equation 9*:

$$I\left(t\right) = g_e\left(t\right)\left(E_e - v\left(t\right)\right) + g_i\left(t\right)\left(E_i - v\left(t\right)\right) - w\left(t\right) \tag{9}$$

where $g_e$, $g_i$, $v$, and $w$ are extracted from the values of the individual neuron found in *Equation 1* at given time step $t$. In essence, this formulation allowed observation of the model neuron state without intervention. We theorized that inclusion of the computed leak current: $g_L\left(E_L - v\right)$, was unnecessary as the ex vivo neurons express this intrinsically. Our network model generated very large synaptic currents (~500 pA), which is unlikely to reflect realistic inputs to CRH$_{\mathrm{PVN}}$ neurons observed in ex vivo preparation (*Salter et al., 2018*; *Sterley et al., 2018*; *Wamsteeker Cusulin et al., 2013*). Thus, we capped the maximum amplitude of PSCs to ±200 pA without changing the timings of PSPs.

## Software

Simulations and modeling were completed using software packages within the Python 3.5 ecosystem. Single neuron and network simulations were completed using the Brian 2 (2.4) package (*Stimberg et al., 2019*). MSE, EMD, and other mathematical operations were computed using modified implementations of the functions available in NumPy (1.20.1) and SciPy (1.6.0) software packages (*Harris et al., 2020*; *Virtanen et al., 2020*). Backend probability inference was completed using the sbi (0.16.0) package (*Gonçalves et al., 2020*). Optimization of fits was performed using the nevergrad (0.4.3) package (*Liu et al., 2020*). Specifically, we utilized a custom portfolio consisting of two-points differential evolution (two-points DE), covariance matrix adaptation evolutionary strategy (CMA-ES), and a random search using scrambled Hammersley sequence (SCR-Hammersly). Axon binary file (ABF) recordings were imported to python using the pyABF package (*Harden, 2020*). Extraction of single-neuron action potentials and other electrophysiological features from intracellular recordings was performed using the ipfx (0.1.1) package (*Gouwens*

*et al., 2019*). A generalized formulation of the fitting code, including the interface between Brian 2, nevergrad, and sbi are publicly available on GitHub (*Ichiyama and Mestern, 2022*). This includes the methodology for comparing EMD between spike trains in 1D and 2D using a sliced Wasserstein implementation.

## Acknowledgements

We thank Dr. Karl Deisseroth (Stanford University) for providing pAAV-EF1a-double floxed-hChR2(H134R)-EYFP-WPRE-HGHpA was a gift from (Addgene plasmid # 20298; http://n2t.net/addgene:20298; RRID:Addgene_20298).

This work was funded through a Discovery Grant (RGPIN-2015–06106) from the Natural Sciences and Engineering Research Council of Canada (NSERC; to W I), a Project Grant (PJT-148707) from the Canadian Institutes of Health Research (CIHR; to W I), BrainsCAN at Western University through the Canada First Research Excellence Fund (CFREF; to W I and L M), and the Compute Canada (to L M). A I was a recipient of an Ontario Graduate Scholarship, SM is a recipient of the Canadian Open Neuroscience Platform studentship. The funders had no role in study design, data collection and interpretation, or the decision to submit the work for publication

## Additional information

### Funding

| Funder | Grant reference number | Author |
|---|---|---|
| Natural Sciences and Engineering Research Council of Canada | RGPIN-2015-06106 | Wataru Inoue |
| Canadian Institutes of Health Research | PJT-148707 | Wataru Inoue |
| Canada First Research Excellence Fund | BrainsCAN Accelerator | Wataru Inoue Lyle Muller |
| Compute Canada | | Lyle Muller |
| Canadian Open Neuroscience Platform | | Samuel Mestern Gabriel B Benigno |
| Vector Institute | Postgraduate Affiliate Program | Gabriel B Benigno |

The funders had no role in study design, data collection and interpretation, or the decision to submit the work for publication.

### Author contributions

Aoi Ichiyama, Conceptualization, Data curation, Formal analysis, Investigation, Methodology, Software, Validation, Visualization, Writing – original draft, Writing – review and editing; Samuel Mestern, Conceptualization, Formal analysis, Investigation, Methodology, Software, Visualization, Writing – original draft; Gabriel B Benigno, Formal analysis, Software, Writing – original draft; Kaela E Scott, Methodology; Brian L Allman, Methodology, Supervision, Writing – review and editing; Lyle Muller, Conceptualization, Funding acquisition, Investigation, Methodology, Software, Supervision, Writing – review and editing; Wataru Inoue, Conceptualization, Funding acquisition, Investigation, Supervision, Validation, Writing – original draft, Writing – review and editing

### Author ORCIDs

Aoi Ichiyama (ID) http://orcid.org/0000-0002-0981-2369
Samuel Mestern (ID) http://orcid.org/0000-0001-5062-6712
Lyle Muller (ID) http://orcid.org/0000-0001-5165-9890
Wataru Inoue (ID) http://orcid.org/0000-0002-2438-5123

### Ethics

All experimental procedures were performed in accordance with the Canadian Council on Animal Care guidelines and approved by the University of Western Ontario Animal Use Subcommittee (AUP: 2018-130).

### Decision letter and Author response

Decision letter https://doi.org/10.7554/eLife.76832.sa1
Author response https://doi.org/10.7554/eLife.76832.sa2

---

## Additional files

### Supplementary files
• Transparent reporting form

### Data availability

All data analyzed in this study are included in the manuscript, figures, and figure-supplement. Data analysis code and source code for figures is available at https://github.com/smestern/ichiyama_2022_code, (copy archived at swh:1:rev:d4d153e44882a0c2a7d0a1ef11193347aa5d6c96). The full list of model parameters are listed in Table 1. Figure source data files contain the numerical data used to generate figures.

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
