## [Editor Report]

This is the first electrophysiological study showing high-quality spike activity in vivo from identified CRH neurons in the hypothalamic paraventricular nucleus. The authors make the surprising observation that CRH neurons exhibit brief high-frequency rhythmic bursts in unstressed mice that are converted to a sustained, single spike firing mode by an acute stressor. Their findings are supported by a computational model that suggests that feedback inhibition may regulate the activity patterns of CRH neurons in distinct states. The work provides a framework for new studies exploring firing characteristics in discrete physiological and emotional states in the hypothalamus and perhaps other regions.

---

## [Decision Letter]

**Decision letter after peer review:**

Thank you for submitting your article "State-dependent activity dynamics of hypothalamic stress effector neurons" for consideration by *eLife*. Your article has been reviewed by 3 peer reviewers, one of whom is a member of our Board of Reviewing Editors, and the evaluation has been overseen by John Huguenard as the Senior Editor. The following individuals involved in the review of your submission have agreed to reveal their identity: Akihiro Yamanaka (Reviewer #2); Jaideep Singh Bains (Reviewer #3).

Essential revisions:

1) If the authors have data on what happens either in vivo or in silico if they stimulate the sciatic nerve for long periods, it would improve the manuscript significantly.

2) Please add a discussion of the Yuan et al. 2019 paper which has opposite results in slices after a stressful experience.

3) Please add a thoughtful discussion of the likely effects of the anesthetic on cell firing properties and of bursting more generally.

4) Please discuss the possible mechanism requiring the long silent period between bursts and the role of H-currents, as well as likely contributions of peptide neuromodulators vs. fast neurotransmitters likely contained in the same neurons.

*Reviewer #1 (Recommendations for the authors):*

1. Previous work from thalamic neurons has demonstrated how burst firing can be increased with hyperpolarization, and this can be accompanied by lower firing rates than in the "up-state" in which the cell fires tonically. Do the authors think that a similar modulation may be occurring in the CRH neurons of PVN as well, as many PVN cells express these channels (e.g. Luther et al., 2002, Luther and Tasker, 2000)? Does their model provide any insights into this question? While the decrease in GABAergic inhibition and increased spike attenuation can lead to "stress-like" firing outputs, might a variety of different mechanisms that lead to hyperpolarization also underlie the actual change in vivo?

2. Work by Yuan et al. 2019, cited here, is not really discussed in the context of stressors. The first page of the discussion argues that burst firing in this cell population has not been characterized, but Yuan et al. have indeed reported it. This earlier work indicates that state changes in CRH cells have been previously identified, but muddies the waters as to how these in vitro data can be reconciled with the present work. Yuan et al. used an ex vivo preparation from control or stressed animals and report that the CRH cells also can show burst behaviors after specific stressful stimuli in vivo. However, the neurons are tonically firing from control animals and bursting when in stressed animals. How do the authors think this work fits into their observations in vivo?

3. Why do the graphs for Figure 3G and Figure 3I look nearly identical? This figure may suggest that the burst behavior hardly contributes to the overall firing rate at all, even under basal conditions.

*Reviewer #2 (Recommendations for the authors):*

1) Although computational remodeling well recreates in vitro-like switch between RB and SS, it is not clear as remodeling parameters the amount of contribution of fast neurotransmitters such as GABA and glutamate and slow neurotransmitter such as neuropeptide, CRH. CRH neurons would release both fast neurotransmitter glutamate and slow and sustained neurotransmitter CRH. Does the computational remodeling consider these parameters?

2) CRH release probability from synaptic vesicle might be different between RB and SS. It is not clear if the authors consider this difference in their computational remodeling.

3) Sciatic nerve stimulation-induced transient firing pattern change from RB and SS around 2-3 sec. However, intrinsic state change representatively shown in Figure 3J looks much slower and lasted longer around ~100 sec. Could the authors reproduce a continuous increase of firing by changing the firing pattern from RB to SS if the sciatic nerve stimulation is repetitively applied for a longer period around 100 sec in vivo or in silico?

4) The authors stated that high-frequency burst firing requires a preceding prolonged silence period. It is not clear what neural mechanism is involved in prolonged silence periods in vivo or in silico.

5) H-current through hyperpolarization-activated cyclic nucleotide-gated (HCN) channels play an important role in the generation of rhythmic burst firing. Do the authors record or consider changing of H-current in the activity state switch between RB and SS.

*Reviewer #3 (Recommendations for the authors):*

1) The neurons that were sensitive to blue light ("light-responsive") had a latency of approximately 7 ms. This is sufficient time for action potential in one cell, release of glutamate and an action potential in another cell. Can the authors provide assurances that blue light is directly activating the cell they are recording? In brain slices, light pulses elicit almost simultaneous action potentials, so the delay here needs to be addressed.

2) What is the advantage of a rhythmic burst mechanism during periods when CRH neuron activity is not required for hormone release? The authors offer one sentence in the discussion, but more is warranted here.

3) There needs to be some discussion on the possible contributions of the anesthetic, urethane on the properties described here. This is an important limitation of the study, as it may increase inhibitory tone in the system and needs to be considered.

---

## [Author Response]

Essential revisions:1) If the authors have data on what happens either in vivo or in silico if they stimulate the sciatic nerve for long periods, it would improve the manuscript significantly.

We thank the reviewer for this comment, which helps us to strengthen our proposal that the firing change from RB to SS underlies both transient and more sustained firing rate increase of CRH_PVN_ neurons.

In a pilot in vivo recording, we applied four different stimulation intensities (0.5, 1, 1.5, and 2 mA) of a brief nerve stimulation that was repeated over multiple trials (5 pulses at 20 Hz, every 15 s): the stimulation pattern is identical to what we reported in Figure 3. Notably, at the highest intensity (2 mA), the firing activity did not completely return to the baseline within the trial duration of 15 s, resulting in a continuous decrease of RB and an increase of SS during the stimulation period. These results demonstrate that nerve stimulation (with high intensity and/or short trial intervals) can result in a continuous change from RB to SS, suggesting that common mechanisms underlie both transient and long-lasting change from RB to SS in vivo. In the current study, we focused to study the transient changes triggered by a lower intensity sciatic nerve stimulation, and multiple trials were averaged to analyse firing activity changes.

We added these results as Figure 3—figure supplement in the revised manuscript. We also added the following sentences in the corresponding Results section (Line 185-189).

“In a representative case, we found that the firing pattern change lasted longer with higher intensity stimulation. Figure 3—figure supplement shows that, with the highest intensity stimulation (2 mA), the firing activity change did not completely return to the baseline within the trial duration, resulting in continuous decrease of RB and increase of SS during the stimulation period.”

*In silico,* our model can produce continuous firing pattern changes from RB to SS by persistently lowering GABAergic inputs to CRH neurons and increasing spike-triggered adaptation current in CRH neurons (Figure 5K). In Figure 6O, we also recreated in vivo-like (smooth) transition between RB and SS by transiently changing the same two parameters. Thus, our model results indicate that the sciatic nerve stimulation in vivo triggered transient and reversible changes in neuronal mechanisms which were modelled by these parameters. Our model is consistent with the observation that repetitive (and/or high intensity) sciatic nerve stimulation in vivo produced long-lasting changes in the firing pattern from RB to SS.

2) Please add a discussion of the Yuan et al. 2019 paper which has opposite results in slices after a stressful experience.

We thank the reviewers for pointing out the importance of discussing “burst firing” in slices reported by Yuan et al. (Yuan et al., 2019). To clarify the relationship between our new data and the observation made by Yuan et al. (2019), we added the following section in the Discussion of the revised manuscript (Line 609-619).

“Interestingly, Yuan et al. recently reported spontaneous “burst firing” of CRH_PVN_ neurons ex vivo using cell-attached recordings in slices prepared from repeatedly stressed mice (Yuan et al., 2019). It should be noted, however, that the intra-burst frequency of this ex vivo “burst firing” was <100 Hz (averaging around 50 Hz), which was different from the in vivo RB (>100 Hz intra-burst frequency) we report here in non-stressed mice. Further, the ex vivo “burst firing” involved stress-induced plasticity that developed over days because it only became evident after 3 daily foot shock, but was absent when mice were not previously stressed, or even after 2 daily foot shock. Thus, our data and those by Yuan et al., (2019) are consistent in that CRH_PVN_ neurons do not generate burst firing (and primarily fire in single spiking (SS)) in slices from previously unstressed mice. Accumulating evidence show that CRH_PVN_ neurons undergo diverse forms of synaptic and intrinsic plasticity in response to acute as well as during chronic stress (Bains et al., 2015; Herman and Tasker, 2016; Matovic et al., 2020; Salter et al., 2018; Yuan et al., 2019). Thus, it is an important direction of future studies to examine the plasticity of CRH_PVN_ neurons’ firing in vivo and their contributions to (mal)adaptations of the stress response.”

3) Please add a thoughtful discussion of the likely effects of the anesthetic on cell firing properties and of bursting more generally.

We agree with the reviewers about the importance of discussing the likely effects of anesthetics. We added a new section “Limitations of the study” in our Discussion and added the following paragraph. (Line 675-692)

“Our in vivo recordings were performed under urethane anesthesia, as was the case for the majority of previous studies using in vivo recordings from the PVN in anesthetized rats (Hamamura et al., 1986; Kannan et al., 1987; Saphier, 1989; Saphier and Feldman, 1990, 1988, 1985). Urethane produces a long-lasting steady level of surgical anesthesia while preserving the subcortical and peripheral neural functions (Maggi and Meli, 1986), making it suitable for long electrophysiological recordings from the hypothalamus. For example, under urethane anesthesia in rabbits, hypothalamic neurons have been shown to readily respond to varieties of stress stimuli including hypoxia, hypercapnia, laud noise and pain (Cross and Silver, 1963). Of particular relevance to our study, sciatic nerve stimulation (under urethane anesthesia) has been shown to effectively elevate blood ACTH levels (Hamamura et al., 1986), validating that the stress-responsiveness of the HPA axis is preserved. However, it should be noted that urethane has also been shown to enhance autonomic (Shimokawa et al., 1998b) and HPA axis activity (Hamstra et al., 1984), likely by enhancing adrenergic inputs to the PVN (Shimokawa et al., 1998a). Considering that (nor)adrenaline has both excitatory and inhibitory effects on PVN neurons (Han et al., 2002; Itoi et al., 1994; Saphier and Feldman, 1991) and that it can potently influence state-switch between bursting and single spiking (Pape and McCormick, 1989), urethane may have affected the baseline firing activities and burst firing properties of CRH_PVN_ neurons in general. Future studies using awake animal recordings are warranted.”

4) Please discuss the possible mechanism requiring the long silent period between bursts and the role of H-currents, as well as likely contributions of peptide neuromodulators vs. fast neurotransmitters likely contained in the same neurons.

Regarding the potential roles of H-currents and the possible mechanisms requiring the long silent period for burst firing, we have revised the following sections of the Discussion.

(Line 585-598)

“In acute slices ex vivo, CRH_PVN_ neurons typically show a SS electrophysiological phenotype, or “regular” spiking, with up to around 50 Hz of spike-to-spike frequency in response to steps of intracellular current injections in mice and rats (Bittar et al., 2019; Jiang et al., 2019; Khan et al., 2011; Luther et al., 2002; Matovic et al., 2020; Sarkar et al., 2011; Wamsteeker Cusulin et al., 2013). Thus, our in vivo recordings of the RB (>100 Hz spike-to-spike frequency) revealed a previously underappreciated repertoire of the firing patterns of CRH_PVN_ neurons. In this regard, CRH_PVN_ neurons are different from thalamic relay neurons that have been extensively studied for their properties to switch between bursting and single (tonic) spiking modes both in vivo and ex vivo (McCormick and Pape, 1990a; Sherman, 2001; Steriade et al., 1993). The difference can be partly explained by the fact that CRH_PVN_ neurons express only modest levels of T-type currents (Luther et al., 2002) and little, if any, H-currents (see Figure 5A): these intrinsic properties play crucial roles in the generation of rhythmic burst firing, and switch between burst and single spiking modes in thalamic relay neurons (McCormick and Pape, 1990a; Sherman, 2001; Steriade et al., 1993).”

(Line 634-645)

“Importantly, the burst firing does not simply occur as a results of current injection patterns but requires unique interaction between the intrinsic properties of CRH_PVN_ neurons and the network inputs. This is supported by the following two controls. First, network inputs, which drove subthreshold membrane potential changes preceding (>200 ms) the burst firing, were necessary because truncated (<200 ms) network currents, which only contain high-frequency synaptic inputs component, failed to elicit the brief bursts in CRH_PVN_ neurons (Figure I-K). Notably, this result corroborates well with in vivo data that RB almost exclusively occur after long (>200 ms) silent period (Figure 4B). Second, magnocellular neurons, a separate class of neuroendocrine neurons of the PVN with different intrinsic properties (Luther et al., 2002), did not fire in brief bursts in response to the same network current injection that elicited brief bursts in CRH_PVN_ neurons (Figure 6M, N). The intrinsic properties key to this burst firing are currently unknown and warrants future investigation.”

For more detailed considerations about H-current and other intrinsic mechanisms that may be involved in the burst generation, please see our Responses to Reviewer 1-Point #1 and Reviewer 2-Point #5 below.

Regarding the potential contributions of CRHergic vs. glutamatergic transmission to the generation of RB, we have made the following changes in the discussion. (Line 556-563)

“Furthermore, transmission from CRH neurons to GABAergic neurons is exclusively modelled by slow CRHergic transmission based on recent experimental data (Jiang et al., 2019; Ramot et al., 2017), but CRH_PVN_ neurons have been shown to release glutamate at some synapses (Füzesi et al., 2016). Despite these simplifications, our model parsimoniously captures the key features of firing behavior in vivo, including the brevity of burst episodes and long IBI (Figure 5E, F). Thus, our simplified model, which is the first network model for CRH_PVN_ neurons to our best knowledge, is a useful first step to understand the spiking behaviors of CRH_PVN_ neuron in a network, and to generate experimentally testable hypotheses for future research.”

Reviewer #1 (Recommendations for the authors):1. Previous work from thalamic neurons has demonstrated how burst firing can be increased with hyperpolarization, and this can be accompanied by lower firing rates than in the "up-state" in which the cell fires tonically. Do the authors think that a similar modulation may be occurring in the CRH neurons of PVN as well, as many PVN cells express these channels (e.g. Luther et al., 2002, Luther and Tasker, 2000)? Does their model provide any insights into this question? While the decrease in GABAergic inhibition and increased spike attenuation can lead to "stress-like" firing outputs, might a variety of different mechanisms that lead to hyperpolarization also underlie the actual change in vivo?

We thank the reviewer for this comment. We agree that changes in membrane potential likely underlie the state change between RB and SS spiking modes in CRH_PVN_ neurons, and there are some similarity with thalamic neurons (McCormick and Pape, 1990a; Sherman, 2001; Steriade et al., 1993). Specifically, we consider that hyperpolarized membrane potential may underly the prolonged silence period preceding the burst firing of CRH_PVN_ neurons in vivo (Figure 4), and hence the maintenance of the RB spiking mode and overall low firing rate. On the other hand, we consider that depolarization of membrane potential likely modulates the intrinsic properties of CRH_PVN_ neurons and diminishes their ability to fire the high-frequency burst. As the reviewer pointed out, the models proposed for thalamic neurons can guide us towards further mechanistic understanding. For example, T-type ca^2+^ channels potentially contribute to the generation of burst firing and their inactivation by membrane depolarization may prevent burst firing during the SS mode. Our consideration is in line with the earlier studies reporting that neuroendocrine parvocellular neurons (putative CRH neurons) express moderate levels of T-type currents (Luther et al., 2002; Luther and Tasker, 2000). However, compared to thalamic neurons, the amount of T-type currents in CRH_PVN_ neurons is moderate, and consequently CRH_PVN_ neurons do not readily fire low-threshold burst spike upon release from hyperpolarization (Luther et al., 2002) (also see Figure 5A of our manuscript confirming the finding by Luther et al. 2002). Along the similar line, we found that square pulse depolarization (from hyperpolarized membrane potential below -80 mV) could not trigger high-frequency (>100 Hz) burst firing ex vivo, which was the key characteristic of in vivo RB (Figure 6G, H). Thus, T-type currents alone is not sufficient to explain the in vivo RB. Our data point to additional intrinsic mechanisms (and their interplay with synaptic inputs) in the generation of the high-frequency burst firing as well as the switch between RB and SS modes.

It should also be noted that there are other important differences between the intrinsic properties of CRH_PVN_ neurons and thalamic neurons. First, unlike thalamic relay neurons (McCormick and Pape, 1990a), CRH_PVN_ neurons do not readily generate recurring rhythmic burst firing upon hyperpolarization in acute slices ex vivo (note that the burst firing reported by Yuan et al. 2019 will be discussed in the next section). Second, this is partly because CRH_PVN_ neurons express little, if any, H-currents (see Figure 5A for the absence of Sag current in response to a hyperpolarizing step): our observation is consistent with previous publications from other groups studying intrinsic properties of CRH_PVN_ neurons in mice (see Figure 6G of (J. S. Kim et al., 2019)) and neuroendocrine parvocellular neurons (putative CRH neurons) in rats (see Figure 1 of (Luther and Tasker, 2000)). In the thalamic neurons, H-currents, together with T-type currents, play key roles in the generation of recurring rhythmic burst firing (McCormick and Pape, 1990a).

Our spiking network model indicated that recurrent GABAergic inputs and spike-triggered adaptation were “sufficient” to drive the spiking state change in silico as well as ex vivo. However, this does not exclude the possibility that different mechanisms, which were not incorporated into our current model, may additionally contribute to, or can independently drive, the high-frequency burst firing and the switch between RB and SS modes. For example, neuromodulators, such as noradrenaline and 5-HT, modulate intrinsic and synaptic properties of CRH_PVN_ neurons (Sunstrum and Inoue, 2018) and excite/inhibit CRH_PVN_ neurons (Mukai et al., 2020). Also, in the thalamic relay neurons, both NA and 5-HT causes depolarization of the thalamic neurons and drive the change from burst firing to tonic firing mode (McCormick and Pape, 1990b; Pape and McCormick, 1989). Thus, neuromodulators may play roles in the firing activity switch partly by changing the membrane potentials of CRH_PVN_ neurons.

We have revised the following section of Discussion related to our response.

(Line 585-598)

“In acute slices ex vivo, CRH_PVN_ neurons typically show a SS electrophysiological phenotype, or “regular” spiking, with up to around 50 Hz of spike-to-spike frequency in response to steps of intracellular current injections in mice and rats (Bittar et al., 2019; Jiang et al., 2019; Khan et al., 2011; Luther et al., 2002; Matovic et al., 2020; Sarkar et al., 2011; Wamsteeker Cusulin et al., 2013). Thus, our in vivo recordings of the RB (>100 Hz spike-to-spike frequency) revealed a previously underappreciated repertoire of the firing patterns of CRH_PVN_ neurons. In this regard, CRH_PVN_ neurons are different from thalamic relay neurons that have been extensively studied for their properties to switch between bursting and single (tonic) spiking modes both in vivo and ex vivo (McCormick and Pape, 1990a; Sherman, 2001; Steriade et al., 1993). The difference can be partly explained by the fact that CRH_PVN_ neurons express only modest levels of T-type currents (Luther et al., 2002) and little, if any, H-currents (see Figure 5A): these intrinsic properties play crucial roles in the generation of rhythmic burst firing, and switch between burst and single spiking modes in thalamic relay neurons (McCormick and Pape, 1990a; Sherman, 2001; Steriade et al., 1993).”

(Line 634-645)

“Importantly, the burst firing does not simply occur as a results of current injection patterns but requires unique interaction between the intrinsic properties of CRH_PVN_ neurons and the network inputs. This is supported by the following two controls. First, network inputs, which drove subthreshold membrane potential changes preceding (>200 ms) the burst firing, were necessary because truncated (<200 ms) network currents, which only contain high-frequency synaptic inputs component, failed to elicit the brief bursts in CRH_PVN_ neurons (Figure I-K). Notably, this result corroborates well with in vivo data that RB almost exclusively occur after long (>200 ms) silent period (Figure 4B). Second, magnocellular neurons, a separate class of neuroendocrine neurons of the PVN with different intrinsic properties (Luther et al., 2002), did not fire in brief bursts in response to the same network current injection that elicited brief bursts in CRH_PVN_ neurons (Figure 6M, N). The intrinsic properties key to this burst firing are currently unknown and warrants future investigation.”

2. Work by Yuan et al. 2019, cited here, is not really discussed in the context of stressors. The first page of the discussion argues that burst firing in this cell population has not been characterized, but Yuan et al. have indeed reported it. This earlier work indicates that state changes in CRH cells have been previously identified, but muddies the waters as to how these in vitro data can be reconciled with the present work. Yuan et al. used an ex vivo preparation from control or stressed animals and report that the CRH cells also can show burst behaviors after specific stressful stimuli in vivo. However, the neurons are tonically firing from control animals and bursting when in stressed animals. How do the authors think this work fits into their observations in vivo?

We thank the reviewer for pointing out the importance of discussing “burst firing” in slices reported by Yuan et al. (Yuan et al., 2019). We agree with the reviewer that our original manuscript lacked clear discussion about the relationship between our observations and those by Yuan et al.: this can generate unnecessary confusion. Our considerations are as follows:

1) Our in vivo rhythmic brief bursts (RB) and the ex vivo “burst firing” by Yuan et al. are two distinct modes of firing/bursting. Specifically, the intra-burst frequency is >100 Hz for our in vivo RB and <100 Hz (averaging around 50 Hz) for Yuan et al. 2019 (see Figure 4E of Yuan et al. 2019).

2) The ex vivo “burst firing” by Yuan et al. resulted from stress-induced plasticity because it became evident only when slices were prepared from mice that were repeatedly stressed (3 daily foot shocks), but it was not evident in slices from previously unstressed mice. Interestingly, a single day or 2 daily foot shocks were not sufficient to induce the plasticity required for the “burst firing” indicating its slow development over the course of a few days of repeated stress (see Figure S3 of Yuan et al. 2019). Yuan et al. showed that the burst firing was diminished by bath application of NMDA receptor antagonist, indicating roles of synaptic plasticity.

3) As the reviewer noted, Yuan et al. observed that CRH_PVN_ neurons fire tonically (using cell-attached recording) in slices from previously unstressed mice. Similar observation (using cell-attached recording) was made by an independent group (Sarkar et al., 2011). In this regard, our observation that CRH_PVN_ neurons do not generate burst firing (and primarily fire in single spiking (SS)) in slices from previously unstressed mice is consistent with Yuan et al.

We added the following paragraph in Discussion to clarify the differences between our in vivo RB and ex vivo burst firing by Yuan et al. (Line 608-619)

“Interestingly, Yuan et al. recently reported spontaneous “burst firing” of CRH_PVN_ neurons ex vivo using cell-attached recordings in slices prepared from repeatedly stressed mice (Yuan et al., 2019). It should be noted, however, that the intra-burst frequency of this ex vivo “burst firing” was <100 Hz (averaging around 50 Hz), which was different from the in vivo RB (>100 Hz intra-burst frequency) we report here in non-stressed mice. Further, the ex vivo “burst firing” involved stress-induced plasticity that developed over days because it only became evident after 3 daily foot shocks, but was absent when mice were not previously stressed, or even after 2 daily foot shocks. Accumulating evidence show that CRH_PVN_ neurons undergo diverse forms of synaptic and intrinsic plasticity in response to acute as well as during chronic stress (Bains et al., 2015; Herman and Tasker, 2016; Matovic et al., 2020; Salter et al., 2018; Yuan et al., 2019). Thus, it is an important direction of future studies to examine the plasticity of CRH_PVN_ neurons’ firing in vivo and their contributions to (mal)adaptations of the stress response.”

We also revised the first paragraph of the discussion as follows (Line 471-475):

“We show that CRH_PVN_ neurons, within intact circuits, can fire distinct brief, high-frequency (>100 Hz) bursts, a firing pattern that has not been observed in ex vivo slice electrophysiology (Bittar et al., 2019; Jiang et al., 2019; Khan et al., 2011; Luther et al., 2002; Matovic et al., 2020; Sarkar et al., 2011; Wamsteeker Cusulin et al., 2013)”.

3. Why do the graphs for Figure 3G and Figure 3I look nearly identical? This figure may suggest that the burst behavior hardly contributes to the overall firing rate at all, even under basal conditions.

As the reviewer pointed out, actions potentials associated with burst firing account for a very small proportion of total spikes. Our explanation is as follows: the burst is very brief (on average 3.1 ± 0.5 spikes/burst) and the burst rate is low even during the baseline period (0.1406 ± 0.1375 Hz), this means that the action potential associated with burst firing is approximately 3.1 x 0.1406 = 0.4359 Hz. Please note that during the baseline period neurons do not always fire in the burst mode and occasionally generate single spiking at relatively high rate (Figure 3K). On the other hand, the average spike rate (including both burst and single spikes) during the baseline period is 3.063 ± 1.646 Hz (Results, line181). Taken together, the proportion of action potentials associated with burst firing is approximately 14% of the total spikes. Accordingly, our paper shows that single spiking primarily contributes to the overall the firing rate of CRH_PVN_ neurons, and that burst firing reflects a state of CRH_PVN_ neurons (which is not evident from overall firing rate) where their firing activity is constrained (Figure 3L, M).

Reviewer #2 (Recommendations for the authors):1) Although computational remodeling well recreates in vitro-like switch between RB and SS, it is not clear as remodeling parameters the amount of contribution of fast neurotransmitters such as GABA and glutamate and slow neurotransmitter such as neuropeptide, CRH. CRH neurons would release both fast neurotransmitter glutamate and slow and sustained neurotransmitter CRH. Does the computational remodeling consider these parameters?

The current computational model exclusively used slow (i.e. CRHergic) excitatory transmission for CRH neuron→GABA neurons (τCRH (ms) = 232.6) modelled as single time constant α synapse (Destexhe et al., 1994), reflecting the slow CRHR1 mediated excitation (Jiang et al., 2019; Ramot et al., 2017). For GABA neurons→CRH neurons, our model uses fast GABAergic transmission (τi (ms) = 20.4). In addition, CRH neurons receive fast glutamatergic transmission from external sources (τe (ms) = 12.5) (Materials and methods and Table 1). The rationale for the parameter choice (i.e. CRH instead of glutamate for CRH neuron→GABA neurons synapse) is that recent studies showed that transmission at CRH neuron→GABA(CRHR1) neurons in the PVN (which form recurrent inhibitory circuit with CRH_PVN_ neurons) is almost exclusively mediated by CRH with little contribution by glutamate (Jiang et al., 2019; Ramot et al., 2017). However, as the reviewer pointed out, CRH neurons are known to release glutamate onto other neuron types (Füzesi et al., 2016): these neurons can potentially form recurrent inhibitory circuit with CRH neurons. Thus, our model results provide a testable hypothesis for future experiments to address the roles of specific neurotransmitters and/or co-release for the firing patterns of CRH_PVN_ neurons.

For the clarification of modelling parameters, we have added the following sentences in the Discussion (Line 556-559).

“Furthermore, transmission from CRH neurons to GABAergic neurons is exclusively modelled by slow CRHergic transmission based on recent experimental data (Jiang et al., 2019; Ramot et al., 2017), but CRH_PVN_ neurons have been shown to release glutamate at some synapses (Füzesi et al., 2016).”

2) CRH release probability from synaptic vesicle might be different between RB and SS. It is not clear if the authors consider this difference in their computational remodeling.

We agree with the reviewer’s comment that different spiking modes (RB and SS) may change the release probability of CRH release. Our current computational model, however, does not incorporate this interesting possibility. This is because experimental data for the relationship between spiking patterns and CRH release is currently not well characterized. Our in vivo data revealing RB and SS spiking modes offers future studies to apply physiologically relevant spiking patterns to examine how spiking patterns affect CRH release. The experimental data, in turn, will further improve the computational model.

3) Sciatic nerve stimulation-induced transient firing pattern change from RB and SS around 2-3 sec. However, intrinsic state change representatively shown in Figure 3J looks much slower and lasted longer around ~100 sec. Could the authors reproduce a continuous increase of firing by changing the firing pattern from RB to SS if the sciatic nerve stimulation is repetitively applied for a longer period around 100 sec in vivo or in silico?

We thank the reviewer for this comment, which helps us to strengthen our proposal that the firing change from RB to SS underlies both transient and more sustained firing rate increase of CRH_PVN_ neurons.

In a pilot in vivo recording, we applied four different stimulation intensities (0.5, 1, 1.5, and 2 mA) of a brief nerve stimulation that was repeated over multiple trials (5 pulses at 20 Hz, every 15 s): the stimulation pattern is identical to what we reported in Figure. 3. Notably, at the highest intensity (2 mA), the firing activity did not completely return to the baseline within the trial duration of 15 s, resulting in continuous decrease of RB and increase of SS during the stimulation period. These results demonstrate that nerve stimulation (with high intensities and/or short trial intervals) can result in continuous change from RB to SS, suggesting that common mechanisms underlies both transient and long-lasting change from RB to SS in vivo. In the current study, we focused to study the transient changes triggered by a lower intensity (1.6 mA) sciatic nerve stimulation, and multiple trials were averaged to analyse firing activity changes.

We added these results as Figure 3—figure supplement. We also added the following sentences in corresponding Result section (Line 185-189).

“In a representative case, we found that the firing pattern change lasted longer with higher intensity stimulation. Figure 3—figure supplement 1 shows that, with the highest intensity stimulation (2 mA), the firing activity change did not completely return to the baseline within the trial duration of 15 s, resulting in continuous decrease of RB and increase of SS during the stimulation period.”

We do not exclude the possibility that long stimulation (with higher intensity) could lead to lasting plasticity and irreversible change in the firing pattern after the cessation of stimulation. However, mechanisms that involved lasting plasticity are beyond the scope of the current study which focuses on the neural mechanisms for rapid, and reversible firing changes.

In silico, our model can produce continuous firing pattern change from RB to SS by persistently lowering GABAergic inputs to CRH neurons and increasing spike-triggered adaptation current in CRH neurons (Figure 5K). In Figure 6O, we also recreated in vivo-like (smooth) transition between RB and SS by transiently changing the same two parameters. Thus, our model results indicate that the sciatic nerve stimulation in vivo triggered transient and reversible changes in nerve mechanisms which were modelled by these parameters. Our model is also consistent with the observation that repetitive (and/or high intensity) sciatic nerve stimulation in vivo produced long-lasting changes in the firing pattern from RB to SS.

4) The authors stated that high-frequency burst firing requires a preceding prolonged silence period. It is not clear what neural mechanism is involved in prolonged silence periods in vivo or in silico.

We thank the reviewer for this comment that helps us to clarify the neural mechanisms involved in prolonged silence.

Our spiking network model shown in Figure 5C recapitulated in vivo-like rhythmic brief bursting (RB) and prolonged silence preceding the high frequency burst firing in model CRH neurons (Figure 5D). Our model indicates that recurrent inhibitory inputs almost exclusively drive the prolonged silence. More specifically, the model CRH neurons receive noisy excitatory inputs at an average frequency of 30 Hz. On the other hand, the recurrent inhibitory inputs are dynamic, driven by the burst firing of CRH neurons → prolonged CRHergic excitation (described in point #1) → prolonged excitation of GABA neurons. As the feedback inhibition decreases (i.e. disinhibition), the constant external excitatory inputs can generate membrane depolarization sufficient to trigger the next high-frequency burst firing. In addition to the inhibitory and excitatory synaptic inputs, the spike-triggered adaptation current, which represents after-hyperpolarization, (a loss of) after-depolarization, or both (Izhikevich, 2003), prevents the action potential firing of CRH neurons after the burst firing. However, its influence is less important. This is because its time constant (τ_w_ = 98.2 ms, Table 1) is substantially shorter than the time constant of CRHergic excitation (τ_CRH_ = 232.6 ms, Table 1) and ensuing prolonged feedback inhibition.

To clarify the contribution of inhibitory inputs to the prolonged silence, we added Figure 5—figure supplement in the revised manuscript that shows the temporal relationship between the burst firing patterns of representative model CRH neurons and the breakdown of three model parameters (i.e. excitatory conductance, inhibitory conductance and adaptation current). We also added the following description in the Results of the revised manuscript (Line 374-382).

“In the model CRH neurons, the recurrent inhibitory inputs were critical for the long, mostly silent IBI, and the loss of inhibitory inputs (i.e. disinhibition) underlay the timing when excitatory inputs generated depolarization sufficient to trigger burst firing (Figure 5—figure supplement). In addition to the inhibitory and excitatory synaptic inputs, the spike-triggered adaptation current (Izhikevich, 2003), prevented the action potential firing of CRH neurons after the burst firing. However, its influence was less important. This was because its time constant (τw = 98.2 ms, Table 1) was substantially shorter than the time constant of CRHergic excitation (τCRH = 232.6 ms, Table 1) and ensuing prolonged feedback inhibition.”

In ex vivo patch-clamp recordings, CRH_PVN_ neurons do not intrinsically generate RB unlike thalamic relay neurons that have been shown to switch between burst and single spiking modes (McCormick and Pape, 1990a; Sherman, 2001; Steriade et al., 1993). Thus, we reasoned that in vivo-like synaptic inputs (as modelled above) play important roles in the rhythmic burst generation in CRH_PVN_ neurons. To this end, we used the current injection protocol derived from our spiking network model. Our data showed that the network-derived current protocol was sufficient to generate the high-frequency burst firing, together with the prolonged silence preceding the burst firing in biological CRH_PVN_ neurons.

Interestingly, the prolonged silence had an important influence on the intrinsic properties of the biological CRH_PVN_ neurons (Figure 6I-K) because injections of truncated network-derived current (which temporary correspond with the timing of burst firing) did not reliably elicit burst firing, and the preceding noisy network inputs (which caused subthreshold membrane potential fluctuations) were necessary to efficiently drive the burst firing (Figure 6I-K). Based on these observations, we speculate that inhibitory feedback inputs play key roles in the prolonged silence/subthreshold membrane potential fluctuations, and consequently the ensuing generation of high-frequency burst firing.

The neural mechanisms for the prolonged silence required for the high-frequency burst firing in vivo would be most directly examined by in vivo patch clamp recordings, which is beyond the scope of this paper and an important direction of future research.

5) H-current through hyperpolarization-activated cyclic nucleotide-gated (HCN) channels play an important role in the generation of rhythmic burst firing. Do the authors record or consider changing of H-current in the activity state switch between RB and SS.

In ex vivo patch-clamp recordings shown in Figure 5A, we observed little, if any, Sag current in CRH_PVN_ neurons in response to a hyperpolarizing current step (from −70 mV to near −100 mV). While the holding potential of −70 mV is not ideal for studying Sag current as it partially opens HCN channels, it is still well above its maximum activation voltage (~−100 mV) (McCormick and Pape, 1990a). Therefore, if CRH_PVN_ neurons possess prominent HCN channels, Sag current should be readily detectable in our recordings. Our observation is consistent with previous publications from other groups that reported similar current clamp trances from CRH_PVN_ neurons in mice (Figure 6G, (J. S. Kim et al., 2019)) and parvocellular neuroendocrine neurons (putative CRH neurons) in rats (Figure 1, (Luther and Tasker, 2000)). Thus, we conclude that HCN channels makes little, if any, contributions to the intrinsic membrane properties of CRH_PVN_ neurons involved in the burst firing.

However, our data do not fully exclude the possibility that HCN channels may become facilitated in CRH_PVN_ neurons, for example after stress, by neuromodulators that increase intracellular cAMP. As the reviewer pointed out, HCN channels play important roles in burst firing, with both facilitatory and inhibitory effects, depending on the magnitude of activity (McCormick and Pape, 1990a). With relevance to the firing mode switch, it has been shown that facilitation of HCN channels by noradrenaline and 5-HT contribute to the switch from rhythmic burst firing to single spike firing in thalamic relay neurons (McCormick and Pape, 1990b). In this case, activation of HCN channels prevented hyperpolarization required for subsequent burst generation. The same study also showed that HCN channels increased after-hyperpolarization and prevented burst firing. In this regard, HCN channels are potential intrinsic properties reflected by the spike-triggered adaptation current in our working model.

Reviewer #3 (Recommendations for the authors):1) The neurons that were sensitive to blue light ("light-responsive") had a latency of approximately 7 ms. This is sufficient time for action potential in one cell, release of glutamate and an action potential in another cell. Can the authors provide assurances that blue light is directly activating the cell they are recording? In brain slices, light pulses elicit almost simultaneous action potentials, so the delay here needs to be addressed.

We appreciate the reviewer’s concern that 7 ms could be sufficient for the excitation of the postsynaptic neurons via fast glutamatergic transmission. For the considerations listed below, we believe that the light-responsive neurons are directly activated by blue light.

1) In the PVN, CRH neurons are unlikely to excite other neurons via fast glutamatergic transmission. Jiang et al. showed that optogenetic stimulation of CRH_PVN_ neurons excites CRHR1 neurons, but this indirect excitation required trains of light stimulation, slow onset and almost exclusively mediated by CRH. Indeed, the optogenetic stimulation of CRH neurons evoked fast EPSCs in only 2% of CRHR1 neurons (Jiang et al., 2018). We validate that our recording electrode hit the PVN with post-mortem histology. Therefore, although we cannot fully exclude the possibility of secondary activation, it is highly unlikely for our optogenetic stimulation to evoke indirect activation after a single light stimulus with a latency of approximately 7 ms.

2) Outside of the PVN, CRH neurons have been shown to release glutamate onto neurons in the lateral hypothalamus (Füzesi et al., 2016). They can be excited indirectly by blue light. However, as described above, our recordings were validated to be made in the PVN: we always found our electrode tract between the 3^rd^ ventricle and the fornix.

3) One key factor for the latency of direct light-triggered action potential is the intensity of the light-induced current in the target neurons, which will be determined by the combination of light intensity and the density of ChR2 expression. Indeed, even in ex vivo, the latency can vary (2-10 ms) within the same neurons depending of the light intensity [Figure 6C of (Wamsteeker Cusulin et al., 2013)]. in vivo, the light intensity (at the target neuron) is difficult to control. We tried to avoid increasing light intensity (kept it just enough to reliably trigger action potentials) because high intensity light tends to generate large light-induced artifacts which prevent the detection of short-latency action potentials.

2) What is the advantage of a rhythmic burst mechanism during periods when CRH neuron activity is not required for hormone release? The authors offer one sentence in the discussion, but more is warranted here.

We thank the reviewer for this comment and for encouraging us to think more about the significance of burst firing. We took the opportunity that *eLife* offers, that is authors are allowed include “Ideas and Speculation” in the Discussion.

We added the following paragraph at the end of Discussion as “Ideas and Speculations”. (Line 694-716)

“Our study found brief, high-frequency burst firing of CRH_PVN_ neurons in vivo. This unexpected finding led us to propose, using a tight combination of experimental and computational approaches, that recurrent inhibition constrains the firing rate of CRH_PVN_ neurons and ensuing hormonal release. Then, what is the advantage of a RB mechanism during periods when CRH neuron activity is not required for hormone release? Does RB firing encode and convey specific information? One possibility is that burst firing facilitates the self-maintenance of the RB firing state. That is, a brief burst enables high fidelity synaptic transmission (Lisman, 1997) and efficiently excites GABAergic neurons that return recurrent inhibition to CRH_PVN_ neurons: this feedback inhibition, in turn, facilitates subsequent burst firing. Notably, our spiking network model found that recurrent inhibition works as a gain regulator at CRH neurons (Figure 5). Thus, our model prediction offers a new circuit mechanism for seminal experimental studies that revealed disinhibition, more so than increasing excitatory inputs, activates CRH_PVN_ neurons for hormonal release (Cole and Sawchenko, 2002; Hewitt et al., 2009; Kovács et al., 2004; Sarkar et al., 2011). Beyond constraining hormonal release of CRH, the idea that CRH_PVN_ neurons use brief bursts for high fidelity synaptic transmission, without triggering massive HPA axis activation, raises an intriguing possibility in light of emerging new roles of CRH_PVN_ neurons. Recent studies showed that CRH_PVN_ neurons are involved in controlling wakefulness (Ono et al., 2020), valence encoding (J. Kim et al., 2019), reward processing (Yuan et al., 2019), and defensive behavior control (Daviu et al., 2020; Füzesi et al., 2016) under non-stress and stress conditions. We propose that burst firing is an ideal mode of signal transmission for CRH_PVN_ neurons to drive rapid behavioral and emotional changes without triggering systemic and long-lasting hormonal response.”

3) There needs to be some discussion on the possible contributions of the anesthetic, urethane on the properties described here. This is an important limitation of the study, as it may increase inhibitory tone in the system and needs to be considered.

We added a new section “Limitations of the study” in our Discussion and added the following paragraph specifically for this topic. (Line 675-692)

“Our in vivo recordings were performed under urethane anesthesia, as was the case for the majority of previous studies using in vivo recordings from the PVN in anesthetized rats (Hamamura et al., 1986; Kannan et al., 1987; Saphier, 1989; Saphier and Feldman, 1990, 1988, 1985). Urethane produces a long-lasting steady level of surgical anesthesia, and preserves the subcortical and peripheral neural functions (Maggi and Meli, 1986), making it suitable for long electrophysiological recordings from the hypothalamus. For example, under urethane anesthesia in rabbits, hypothalamic neurons have been shown to readily respond to varieties of stress stimuli including hypoxia, hypercapnia, laud noise and pain (Cross and Silver, 1963). Of particular relevance to our study, sciatic nerve stimulation (under urethane anesthesia) has been shown to effectively elevate blood ACTH levels (Hamamura et al., 1986), validating that the stress-responsiveness of the HPA axis is preserved. However, it should be noted that urethane has also been shown to enhance autonomic (Shimokawa et al., 1998b) and HPA axis activity (Hamstra et al., 1984), likely by enhancing adrenergic inputs to the PVN (Shimokawa et al., 1998a). Considering that (nor)adrenaline has both excitatory and inhibitory effects on PVN neurons (Han et al., 2002; Itoi et al., 1994; Saphier and Feldman, 1991) and that it can potently influence state-switch between bursting and single spiking (Pape and McCormick, 1989), urethane may have affected the baseline firing activities and burst firing properties of CRH_PVN_ neurons in general. Future studies using awake animal recordings are warranted.”

References:

Bittar TP, Nair BB, Kim JS, Chandrasekera D, Sherrington A, Iremonger KJ. 2019. Corticosterone mediated functional and structural plasticity in corticotropin-releasing hormone neurons. Neuropharmacology. doi:10.1016/j.neuropharm.2019.02.017

Cole RL, Sawchenko PE. 2002. Neurotransmitter Regulation of Cellular Activation and Neuropeptide Gene Expression in the Paraventricular Nucleus of the Hypothalamus. J Neurosci 22:959–969.

Cross BA, Silver IA. 1963. Unit activity in the hypothalamus and the sympathetic response to hypoxia and hypercapnia. Experimental Neurology 7:375–393. doi:10.1016/0014-4886(63)90019-0

Daviu N, Füzesi T, Rosenegger DG, Rasiah NP, Sterley T-L, Peringod G, Bains JS. 2020. Paraventricular nucleus CRH neurons encode stress controllability and regulate defensive behavior selection. Nat Neurosci 23:398–410. doi:10.1038/s41593-020-0591-0

Destexhe A, Mainen ZF, Sejnowski TJ. 1994. Synthesis of models for excitable membranes, synaptic transmission and neuromodulation using a common kinetic formalism. J Comput Neurosci 1:195–230. doi:10.1007/BF00961734

Füzesi T, Daviu N, Wamsteeker Cusulin JI, Bonin RP, Bains JS. 2016. Hypothalamic CRH neurons orchestrate complex behaviours after stress. Nature Communications 7:11937. doi:10.1038/ncomms11937

Hamamura M, Onaka T, Yagi K. 1986. Parvocellular neurosecretory neurons: converging inputs after saphenous nerve and hypovolemic stimulations in the rat. The Japanese journal of physiology 36:921–933. doi:10.2170/jjphysiol.36.921

Hamstra WN, Doray D, Dunn JD. 1984. The effect of urethane on pituitary-adrenal function of female rats. Acta Endocrinol (Copenh) 106:362–367. doi:10.1530/acta.0.1060362

Han SK, Chong W, Li LH, Lee IS, Murase K, Ryu PD. 2002. Noradrenaline excites and inhibits GABAergic transmission in parvocellular neurons of rat hypothalamic paraventricular nucleus. J Neurophysiol 87:2287–2296.

Hewitt SA, Wamsteeker JI, Kurz EU, Bains JS. 2009. Altered chloride homeostasis removes synaptic inhibitory constraint of the stress axis. Nat Neurosci 12:438–443. doi:10.1038/nn.2274

Itoi K, Suda T, Tozawa F, Dobashi I, Ohmori N, Sakai Y, Abe K, Demura H. 1994. Microinjection of norepinephrine into the paraventricular nucleus of the hypothalamus stimulates corticotropin-releasing factor gene expression in conscious rats. Endocrinology 135:2177–2182. doi:10.1210/en.135.5.2177

Izhikevich EM. 2003. Simple model of spiking neurons. IEEE Transactions on Neural Networks 14:1569–1572. doi:10.1109/TNN.2003.820440

Jiang Z, Rajamanickam S, Justice NJ. 2019. CRF signaling between neurons in the paraventricular nucleus of the hypothalamus (PVN) coordinates stress responses. Neurobiol Stress 11. doi:10.1016/j.ynstr.2019.100192

Kannan H, Kasai M, Osaka T, Yamashita H. 1987. Neurons in the paraventricular nucleus projecting to the median eminence: a study of their afferent connections from peripheral baroreceptors, and from the A1-catecholaminergic area in the ventrolateral medulla. Brain Res 409:358–363. doi:10.1016/0006-8993(87)90722-0

Khan AM, Kaminski KL, Sanchez-Watts G, Ponzio TA, Kuzmiski JB, Bains JS, Watts AG. 2011. MAP kinases couple hindbrain-derived catecholamine signals to hypothalamic adrenocortical control mechanisms during glycemia-related challenges. J Neurosci 31:18479–18491. doi:10.1523/JNEUROSCI.4785-11.2011

Kim J, Lee S, Fang Y-Y, Shin A, Park S, Hashikawa K, Bhat S, Kim D, Sohn J-W, Lin D, Suh GSB. 2019. Rapid, biphasic CRF neuronal responses encode positive and negative valence. Nat Neurosci 22:576–585. doi:10.1038/s41593-019-0342-2

Kim JS, Han SY, Iremonger KJ. 2019. Stress experience and hormone feedback tune distinct components of hypothalamic CRH neuron activity. Nature Communications 10:5696. doi:10.1038/s41467-019-13639-8

Kovács KJ, Miklós IH, Bali B. 2004. GABAergic mechanisms constraining the activity of the hypothalamo-pituitary-adrenocortical axis. Ann N Y Acad Sci 1018:466–476. doi:10.1196/annals.1296.057

Lisman JE. 1997. Bursts as a unit of neural information: making unreliable synapses reliable. Trends Neurosci 20:38–43. doi:10.1016/S0166-2236(96)10070-9

Luther JA, Daftary SS, Boudaba C, Gould GC, Halmos KC, Tasker JG. 2002. Neurosecretory and non-neurosecretory parvocellular neurones of the hypothalamic paraventricular nucleus express distinct electrophysiological properties. Journal of Neuroendocrinology 14:929–932. doi:10.1046/j.1365-2826.2002.00867.x

Luther JA, Tasker JG. 2000. Voltage-gated currents distinguish parvocellular from magnocellular neurones in the rat hypothalamic paraventricular nucleus. J Physiol 523:193–209. doi:10.1111/j.1469-7793.2000.t01-1-00193.x

Maggi CA, Meli A. 1986. Suitability of urethane anesthesia for physiopharmacological investigations in various systems Part 1: General considerations. Experientia 42:109–114. doi:10.1007/BF01952426

Matovic S, Ichiyama A, Igarashi H, Salter EW, Sunstrum JK, Wang XF, Henry M, Kuebler ES, Vernoux N, Martinez‐Trujillo J, Tremblay M-E, Inoue W. 2020. Neuronal hypertrophy dampens neuronal intrinsic excitability and stress responsiveness during chronic stress. The Journal of Physiology 598:2757–2773. doi:10.1113/JP279666

McCormick DA, Pape HC. 1990a. Properties of a hyperpolarization-activated cation current and its role in rhythmic oscillation in thalamic relay neurones. J Physiol 431:291–318. doi:10.1113/jphysiol.1990.sp018331

McCormick DA, Pape HC. 1990b. Noradrenergic and serotonergic modulation of a hyperpolarization-activated cation current in thalamic relay neurones. J Physiol 431:319–342. doi:10.1113/jphysiol.1990.sp018332

Mukai Y, Nagayama A, Itoi K, Yamanaka A. 2020. Identification of substances which regulate activity of corticotropin-releasing factor-producing neurons in the paraventricular nucleus of the hypothalamus. Scientific Reports 10:13639. doi:10.1038/s41598-020-70481-5

Ono D, Mukai Y, Hung CJ, Chowdhury S, Sugiyama T, Yamanaka A. 2020. The mammalian circadian pacemaker regulates wakefulness via CRF neurons in the paraventricular nucleus of the hypothalamus. Science Advances 6:eabd0384. doi:10.1126/sciadv.abd0384

Pape HC, McCormick DA. 1989. Noradrenaline and serotonin selectively modulate thalamic burst firing by enhancing a hyperpolarization-activated cation current. Nature 340:715–718. doi:10.1038/340715a0

Ramot A, Jiang Z, Tian J-B, Nahum T, Kuperman Y, Justice N, Chen A. 2017. Hypothalamic CRFR1 is essential for HPA axis regulation following chronic stress. Nat Neurosci 20:385–388. doi:10.1038/nn.4491

Saphier D. 1989. Catecholaminergic projections to tuberoinfundibular neurones of the paraventricular nucleus: I. Effects of stimulation of A1, A2, A6 and C2 cell groups. Brain Research Bulletin 23:389–395. doi:10.1016/0361-9230(89)90179-2

Saphier D, Feldman S. 1991. Catecholaminergic projections to tuberoinfundibular neurones of the paraventricular nucleus: III. Effects of adrenoceptor agonists and antagonists. Brain Res Bull 26:863–870.

Saphier D, Feldman S. 1990. Iontophoresis of cortisol inhibits responses of identified paraventricular nucleus neurones to sciatic nerve stimulation. Brain Res 535:159–162.

Saphier D, Feldman S. 1988. Iontophoretic application of glucocorticoids inhibits identified neurones in the rat paraventricular nucleus. Brain Res 453:183–190.

Saphier D, Feldman S. 1985. Effects of neural stimuli on paraventricular nucleus neurones. Brain Res Bull 14:401–407. doi:10.1016/0361-9230(85)90016-4

Sarkar J, Wakefield S, MacKenzie G, Moss SJ, Maguire J. 2011. Neurosteroidogenesis is required for the physiological response to stress: role of neurosteroid-sensitive GABAA receptors. J Neurosci 31:18198–18210. doi:10.1523/JNEUROSCI.2560-11.2011

Sherman SM. 2001. Tonic and burst firing: dual modes of thalamocortical relay. Trends Neurosci 24:122–126. doi:10.1016/s0166-2236(00)01714-8

Shimokawa, Jin QH, Ishizuka Y, Kunitake T, Takasaki M, Kannan H. 1998a. Effects of anesthetics on norepinephrine release in the hypothalamic paraventricular nucleus region of awake rats. Neurosci Lett 244:21–24. doi:10.1016/s0304-3940(98)00119-0

Shimokawa, Kunitake T, Takasaki M, Kannan H. 1998b. Differential effects of anesthetics on sympathetic nerve activity and arterial baroreceptor reflex in chronically instrumented rats. J Auton Nerv Syst 72:46–54. doi:10.1016/s0165-1838(98)00084-8

Steriade M, McCormick DA, Sejnowski TJ. 1993. Thalamocortical oscillations in the sleeping and aroused brain. Science 262:679–685. doi:10.1126/science.8235588

Sunstrum JK, Inoue W. 2018. Heterosynaptic modulation in the paraventricular nucleus of the hypothalamus. Neuropharmacology. doi:10.1016/j.neuropharm.2018.11.004

Wamsteeker Cusulin JI, Füzesi T, Watts AG, Bains JS. 2013. Characterization of corticotropin-releasing hormone neurons in the paraventricular nucleus of the hypothalamus of Crh-IRES-Cre mutant mice. PLoS ONE 8:e64943. doi:10.1371/journal.pone.0064943

Yuan Y, Wu W, Chen M, Cai F, Fan C, Shen W, Sun W, Hu J. 2019. Reward Inhibits Paraventricular CRH Neurons to Relieve Stress. Curr Biol 29:1243-1251.e4. doi:10.1016/j.cub.2019.02.048